# GRAPHROUTER: A GRAPH-BASED ROUTER FOR LLM SELECTIONS

**Tao Feng, Yanzhen Shen, Jiaxuan You**
Department of Computer Science
University of Illinois Urbana Champaign Urbana, IL, USA
`{taofeng2, yanzhen4, jiaxuan}@illinois.edu`

## ABSTRACT

The rapidly growing number and variety of Large Language Models (LLMs) present significant challenges in efficiently selecting the appropriate LLM for a given query, especially considering the trade-offs between performance and computational cost. Current LLM selection methods often struggle to generalize across new LLMs and different tasks because of their limited ability to leverage contextual interactions among tasks, queries, and LLMs, as well as their dependence on a transductive learning framework. To address these shortcomings, we introduce a novel inductive graph framework, named as `GraphRouter`, which fully utilizes the contextual information among tasks, queries, and LLMs to enhance the LLM selection process. `GraphRouter` constructs a heterogeneous graph comprising task, query, and LLM nodes, with interactions represented as edges, which efficiently captures the contextual information between the query's requirements and the LLM's capabilities. Through an innovative edge prediction mechanism, `GraphRouter` is able to predict attributes (the effect and cost of LLM response) of potential edges, allowing for optimized recommendations that adapt to both existing and newly introduced LLMs without requiring retraining. Comprehensive experiments across three distinct effect-cost weight scenarios have shown that `GraphRouter` substantially surpasses existing routers, delivering a minimum performance improvement of 12.3%. In addition, it achieves enhanced generalization across new LLMs settings and supports diverse tasks with at least a 9.5% boost in effect and a significant reduction in computational demands. This work endeavors to apply a graph-based approach for the contextual and adaptive selection of LLMs, offering insights for real-world applications. Our codes for `GraphRouter` is released at https://github.com/ulab-uiuc/GraphRouter.

## 1 INTRODUCTION

The field of Large Language Models (LLMs) is advancing quickly, offering an increasingly diverse range of models that vary in size, functionality, and computational demands (Bang, 2023; Liu et al., 2023). Although larger models tend to deliver better performance, their high computational costs make them inefficient for many less complex tasks (Snell et al., 2024; Chen & Varoquaux, 2024). Additionally, LLMs demonstrate varied performance across different types of queries and tasks (Ahmed et al., 2024; Zhang et al., 2024), especially with the development of domain-specific LLMs (Singhal et al., 2022; Luo et al., 2022). These challenges make it difficult to recommend the optimal LLM services to users that strike the balance between performance and cost for their specific needs. Therefore, our paper aims to raise attention to this pressing research question: *Given the vast and continuously evolving landscape of LLMs, how to recommend appropriate LLMs for various user queries with different implied tasks?*

Existing researchers have proposed to develop a router to assign a specific LLM to each user query. Hybrid LLM (Ding et al., 2024) trains a binary score router function to determine whether to select a small LLM or a large LLM for a specific query. Although it balances cost and performance, it is limited to just two LLMs, which falls short of addressing the real-world demand for a wide range of LLMs. Some other studies (Dai et al., 2024; Chen et al., 2023) have introduced more advanced router models to address the challenge of selecting among a limited set of LLMs (typically 3 to 5). More

Table 1: **Comparison of `GraphRouter` with existing methods from three perspectives: contextual info, generalization to new LLMs, and multi-task support.** Compared to other approaches, `GraphRouter` introduces an inductive graph framework that fully leverages contextual information, enabling it to generalize to new LLMs and adapt to a variety of tasks.

| Method | Contextual Info | Generalization to New LLMs | Multi-task Support |
|---|---|---|---|
| Hybrid LLM (Ding et al., 2024) | Indices | ✗ | ✗ |
| FrugalGPT (Chen et al., 2023) | LLM name | ✗ | ✗ |
| C2MAB-V (Dai et al., 2024) | One-hot embedding | ✗ | ✗ |
| GraphRouter | Graph-based contexts | ✓ | ✓ |

specifically, FrugalGPT (Chen et al., 2023) proposes a router model based on BERT (Devlin, 2018) to determine whether to switch to a larger LLM or not, and C2MAB-V (Dai et al., 2024) constructs a bandit-based router to balance between the exploration and exploitation when choosing LLM for the user. However, as shown in Table 1, they are still limited in the following aspects: 1) Relying solely on basic BERT-based embeddings to distinguish queries, and on names or indices to distinguish LLMs, they fail to fully leverage the contextual information from the interaction between the task, query, and LLM. This makes it challenging to achieve a router with strong generalization capabilities. 2) These methods rely on a *transductive* learning framework (Arnold et al., 2007; Joachims, 2003), which makes them ill-suited for real-world applications where new LLMs are frequently introduced. When new LLMs are presented, these approaches require retraining with few-shot interaction data before they can be used – a process that is impractical for recommending LLMs to a large number of users in real-time; 3) They train a dedicated router for each specific task, greatly increases the computational overhead and complexity in real-world applications when multiple tasks are present.

To address these challenges, we introduce `GraphRouter`, a graph-based router for LLM selection. `GraphRouter` utilizes an inductive graph framework to effectively leverage contextual information, allowing it to generalize to new LLMs and adapt to diverse tasks. Specifically, to fully utilize the contextual information for different queries and tasks, `GraphRouter` constructs a heterogeneous graph that contains three types of nodes: task node, query node and LLM node. The interaction information between them is represented as edges in a graph. For instance, the reward (including performance and cost) of an LLM responding to a query is modeled as an edge between the query node and the LLM node. Then, we are able to transform the task of predicting the cost and performance of an LLM-query pair to an edge prediction task. After forecasting the properties of the edges, we recommend the most suitable LLM to the user based on their preferences for performance and cost. In addition, in real-world scenarios, new LLMs are frequently developed, so an effective framework should also have the ability to accommodate these evolving models. In order to make `GraphRouter` generalizable to new LLMs, we make efforts in two key aspects. For the input, we utilize a generative LLM such as GPT-4o to generate a descriptive text for each LLM, outlining key details such as its strengths, token pricing, and context length. Based on this, we derive an initial embedding for each LLM using a moderate-size pre-trained language model (e,g, BERT (Devlin, 2018)). This approach offers an advantage over directly using one-hot encoding, as it enables us to generate inductive and more informative initial embeddings for new LLMs. For the `GraphRouter` model, we further develop a heterogeneous GNN that aggregates information from neighboring nodes of different types; given few-shot data, we verified that a trained `GraphRouter` can generalize to new LLM nodes without retraining.

In summary, our main contributions are as follows:

- To the best of our knowledge, we are the first work to build router for LLM selections from the graph perspective, which gives new insight to graph-enhanced LLM research.

- We propose an inductive graph framework that fully leverages contextual information among tasks, queries, and LLMs, enabling it to generalize to new LLMs and adapt to a variety of tasks without retraining.

- In three experiment settings with different performance and cost tradeoffs, `GraphRouter` outperformed the baseline models by at least 12.3%. Furthermore, in scenarios where new LLMs are introduced in the testing data, our method not only saves significant training time but also improves performance by at least 9.5% compared to the baselines.

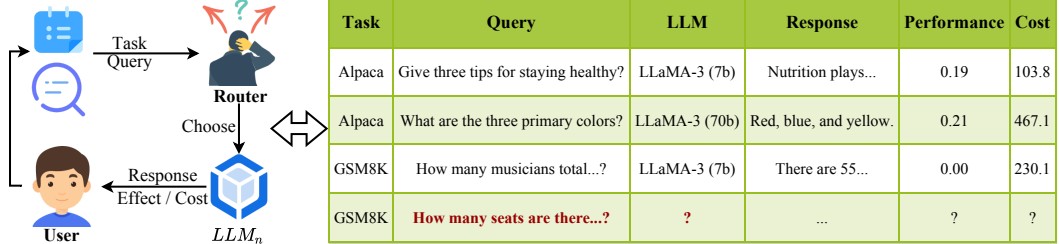

Figure 1: **Overview of `GraphRouter`'s LLM selection process.** As depicted in the left section, the LLM selection process begins with the user inputting a query that belongs to a certain task. When the router receives the query and the task, it will analyze the input and choose the most appropriate $LLM_n$ for generation. Then, $LLM_n$ is used to generate the response. In the end, this response, along with the measured effectiveness and cost, is returned to the user. The right side of the figure illustrates example interaction records, which contain contextual information like task, user query, selected LLM, response, performance, and cost, in a table. These contextualized data are then utilized to train the router.

## 2 GRAPHROUTER: GRAPH-BASED ROUTER FOR LLM SELECTION

### 2.1 PRELIMINARIES

We introduce the LLM selection problem in this section. As shown in the left part of Figure 1, the process involves multiple steps, with the router being the most critical component. The router first receives the user query containing task information. Its goal is to choose an appropriate LLM based on the incoming information in the user query to ensure optimal performance and minimal cost (LLM API cost). After calculation, the router chooses a suitable $LLM_n$ to answer the user query. Finally, the response is returned to the user with its performance and cost. Such an interaction process generates rich contextual data, which contains information on tasks, queries, selected LLM, response, performance, and cost. We organize the data in a table, as shown on the right of Figure 1.

### 2.2 MOTIVATING EXAMPLES

Traditional LLM selection methods (Ding et al., 2024; Chen et al., 2023; Dai et al., 2024) often only use ID or name information to model LLM information, which does not effectively utilize the contextual information (introduced in Sec 2.1) generated from the interaction between LLM and query. Here, we use some examples to illustrate the importance of contextual information. (1) We first argue that contextual information is important because it captures the variance in the ability of different LLMs to respond to diverse queries. We can first observe from Figure 3 that the performance of different LLMs in response to queries can differ significantly. Therefore, understanding the performance patterns of how LLMs handle queries is crucial for LLM selection, and these patterns are embedded in the contextual information between the queries and the LLMs. In addition, from Figure 2, we can also observe that smaller LLMs sometimes outperform larger LLMs on certain queries. Even if we have unlimited budgets and can blindly rely on the largest LLM at a high cost, we still may not achieve optimal performance. This also emphasizes the importance of capturing the varying capabilities of LLMs in handling queries based on contextual information. (2) We also claim that the importance of contextual information lies in its ability to capture the differences in how a single LLM responds to queries across different tasks. Through Figures 3 and 4, we can observe that certain LLMs exhibit significant differences in their performance across two different tasks, such as Mixtral-8x7B (Jiang et al., 2024). These two examples indicate that the performance may vary greatly on different LLMs and tasks. This suggests that in addition to understanding the capabilities of LLMs, the router must also understand the differences and similarities of each task. However, as these critical attributes of LLMs and tasks could not be sufficiently represented by their names or IDs, an effective use of the contextual information that encompasses the interaction between task, query, and LLM is needed.

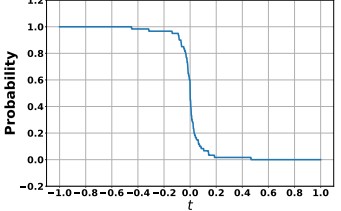

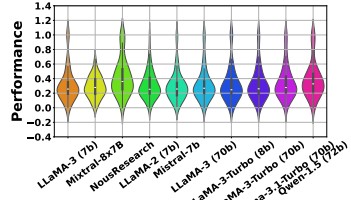

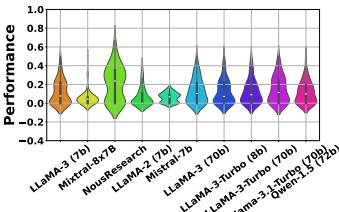

Figure 2: **The probability distribution of a small LLM (LLaMA-3 (7b)) having a better performance value than a large LLM (LLaMA-3 (70b)) by $t$ on the Alpaca dataset, where $t$ means the difference in performance between the small LLM and the large LLM and $t \in [-1, 1]$.**

Figure 3: **Distribution of the performance of different LLMs in response to queries on the Alpaca task**. Specifically, we present a violin plot illustrating the performance of ten LLMs of varying sizes and the dot in each distribution is the median performance.

Figure 4: **Distribution of the performance of different LLMs responding to queries on the SQUAD task**. In particular, the performance of ten LLMs of varying sizes is displayed in a violin plot and the dot in each distribution is the median performance.

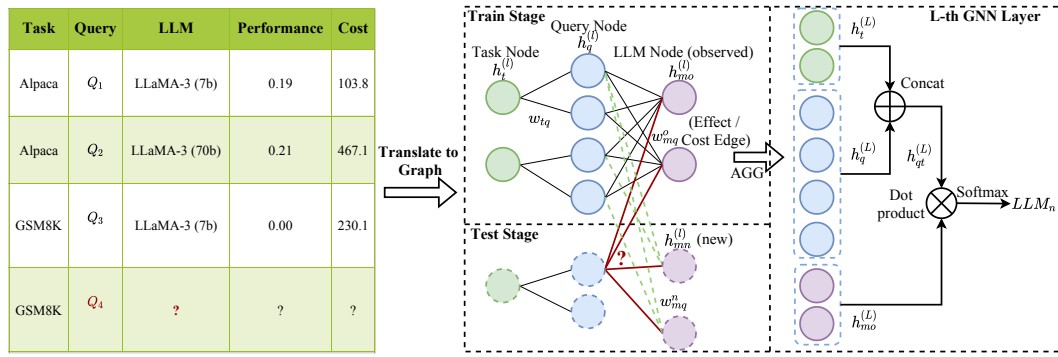

Figure 5: **Overview of `GraphRouter` methodology**. `GraphRouter` first converts the interaction data among tasks, queries, and LLMs into a graph. Specifically, as illustrated on the right side, tasks, queries, and LLMs from the left table are represented as task nodes, query nodes, and LLM nodes, respectively. Moreover, their relationships derived from the interaction data are modeled as edge features. With this structure, we leverage a GNN to embed both node and edge features, ultimately producing the probability distribution of the selected LLM.

## 2.3 GRAPHROUTER FRAMEWORK

**Method Overview.** As shown in Figure 5, `GraphRouter` first transforms the interaction data among tasks, queries, and LLMs into a graph. Specifically, as shown in the right side of Figure 5, we model tasks, queries, and LLMs in the left table as task nodes, query nodes, and LLM nodes, while the relationships derived from the interaction data are represented by edge features. We apply GNN to embed the node and edge features and use them for training and testing.

**Initialize node/edge features.** As shown in Figure 5, we have three types of nodes (task node $h_t^{(l)}$, query node $h_q^{(l)}$, and LLM node $h_m^{(l)}$) and two types of edges (task-query edge $w_{tq}$ and LLM-query edge $w_{mq}$).

Given the inherent differences of task, query, and LLMs, during the initialization of their nodes, we adopt different strategies. For the initialization of task nodes, we utilize an additional LLM, such as GPT-4o, to generate descriptions of the tasks, and then encode the description to obtain its embedding $e_t$. More specifically, we take the average output token embedding after feeding the description into the moderate-size pre-trained language model (PLM), such as BERT (Devlin, 2018). The initialization of query nodes is also obtained by embedding the query $e_q$ through the same PLM. As for the initialization of LLM nodes, the traditional approach often initializes directly using the name or ID of the LLM, which not only limits its ability to generalize to new LLMs but also omits important background information. Here, we still adopt a prompt-based approach. We design prompts

for an LLM to describe the capabilities of each LLM. In addition, we also add information about the cost of each LLM after the description. Then, similar to how we obtain the task's embedding, we use the same PLM to compute the initial embeddings $e_l$ of the different LLMs. All the descriptions we generate for different tasks and LLMs can be found in Appendix A.

As for the task-query edges, we assign a value of **1** for their initialization. For the initialization of LLM-query edges, we jointly consider the performance and cost information in the interaction data, and assign the concatenation of performance and cost as their initial features.

**Predict via a heterogeneous GNN.** We implement the predictive model $f_\phi$ over task nodes, query nodes, and LLM nodes using a heterogeneous GNN, as shown in Figure 5. For aggregating different types of nodes and edges, we use heterogeneous aggregation with different learnable weights. The objective of the GNN is to learn expressive node embeddings $\mathbf{h}$ through an iterative weighted aggregation of the local network neighborhoods. The $l$-th iteration of the GraphConv($\cdot$), or the node embeddings update of the $l$-th layer, is represented as:

$$\mathbf{h}_t^{(l)} = \mathbf{U}_t^{(l)} \text{Concat}\Big( \text{Mean}\big( \{\text{ReLU}(\mathbf{W}_t^{(l)}\mathbf{h}_q^{(l-1)}), q \in \mathcal{N}(t)\} \big), \mathbf{h}_t^{(l-1)} \Big), \quad (1)$$

$$\mathbf{h}_q^{(l)} = \mathbf{U}_q^{(l)} \text{Concat}\Big( \text{Mean}\big( \{\text{ReLU}(\mathbf{w}_{\mathbf{1}[t \in V_t, m \in V_t]}\mathbf{W}_{\mathbf{1}[t \in V_d, m \in V_t]}^{(l)}\mathbf{h}_u^{(l-1)}, u \in \mathcal{N}(v)\} \big), \mathbf{h}_q^{(l-1)} \Big), \quad (2)$$

$$\mathbf{h}_m^{(l)} = \mathbf{U}_m^{(l)} \text{Concat}\Big( \text{Mean}\big( \{\text{ReLU}(\mathbf{w}_{mq}^T\mathbf{W}_m^{(l)}\mathbf{h}_q^{(l-1)}), q \in \mathcal{N}(m)\} \big), \mathbf{h}_m^{(l-1)} \Big), \quad (3)$$

where $\mathbf{h}^{(l)}$ is the node embedding after $l$ iterations, $\mathbf{h}_t^{(l)}, \mathbf{h}_q^{(l)}, \mathbf{h}_m^{(l)}$ have been initialized as $\mathbf{h}_t^{(0)}, \mathbf{h}_q^{(0)}, \mathbf{h}_m^{(0)} = e_t, e_q, e_m$ respectively as explained above, and $\mathcal{N}(v)$ denotes the direct neighbors of $v$. $\mathbf{1}[v \in V_d, u \in V_t]$ indicates the message type (whether from task to query, or LLM to query), and $\mathbf{U}^{(l)}, \mathbf{W}^{(l)}$ are learnable parameters. In addition, $\mathbf{w}_{\mathbf{1}[t \in V_t, m \in V_t]}$ indicates that different edge types correspond to different edge features; specifically, if it is from task to query, it is represented as $w_{tq}$, and from LLM to query, it is represented as $w_{mq}$.

Following the update of the task, query, and LLM node embeddings, we obtain the query-task combined embedding as $\mathbf{h}_{qt}^{(l)} = \text{MLP}(\text{Concat}(\mathbf{h}_t^{(l)}, \mathbf{h}_q^{(l)}))$. We model the LLM selection problem as an edge prediction problem and generate training data in the following way. We first determine the best LLM for each query in the training set based on the performance (best reward described in Sec 3.4) achieved by different LLMs and then set the edge labels of the query to other LLMs to 0 and the edge label of the query to the best LLM to 1. As such, LLM prediction can be made through EdgePred($\cdot$) in the form of $\hat{y}_{logits} = \text{Mean}\Big( \text{Dot}(\mathbf{h}_{qt}^{(l)}, \mathbf{h}_m^{(l)}) \Big)$. We have summarized the detailed training process of `GraphRouter` in Algorithm 1, whose details are shown as follows. In addition, in the testing of `GraphRouter`, we identify the LLM node that has maximum edge logits with the query node as the best LLM, which can be computed as

$$\hat{y} = \arg \max_m \Big( \text{EdgePred}(h_{qt}, h_m) \Big). \quad (4)$$

**`GraphRouter` for new LLMs setting.** Traditional routers cannot generalize to new LLMs directly under few-shots settings, as they require retraining through interactions with the query for each new LLM. This is inadequate for keeping up with the rapidly evolving changes in LLMs in the real world. To test our framework and baselines under this real-world setting, following (Cao et al., 2023; Fey et al., 2023), we construct an auxiliary dataset with the interaction data of the new LLMs on queries sampled uniformly from the training set. This auxiliary dataset is not involved in the training process but serves as a few-shot examples during the testing phase.

## 3 EXPERIMENTAL SETUP

### 3.1 DATASETS AND LLM DESCRIPTIONS

We select data from four different types of tasks, whose statistics are summarized in Table 2.

---

**Algorithm 1** Training of `GraphRouter`

---

**Require:** Dataset $\mathcal{D}_{\text{train}} = \{(\mathbf{x}, y)\}$. A parameterized heterogeneous GNN $f_\phi$. Task-query edge weights $\mathbf{w}_{tq}$ and LLM-query edge weights $\mathbf{w}_{mq}$. Number of GNN layers $L$.

1: Initialize the embeddings of task nodes, query nodes, and LLM nodes, $h_t^{(0)}, h_q^{(0)}, h_m^{(0)}$, using PLM.
2: **for** each iteration $i$ **do**
3:     $M \leftarrow$ SampleMiniEdgeBatch($\mathcal{D}_{\text{train}}$)
4:     Mask the edges in $\mathcal{D}_{\text{train}}$ that are in $M$ and obtain the labels of the edges in $T_{\text{m}}^{(i)}$
5:     **for** $l = 1$ to $L$ **do**
6:         $\mathbf{h}_t^{(l)}, \mathbf{h}_q^{(l)}, \mathbf{h}_m^{(l)} \leftarrow$ GraphConv($\mathbf{h}_t^{(l-1)}, \mathbf{h}_q^{(l-1)}, \mathbf{h}_m^{(l-1)}, \mathbf{w}_{tq}, \mathbf{w}_{mq}$) with $f_\phi$
7:     $\hat{y}_{logits} \leftarrow$ EdgePred($\mathbf{h}_t^{(l)}, \mathbf{h}_q^{(l)}), \mathbf{h}_m^{(l)}$ with $f_\phi$
8:     Backward $\left(\text{Criterion}(\hat{y}_{logits}, \{\{y_j\}_{j \in T_{\text{m}}^{(i)}}\} \in M)\right)$

---

- **Alpaca** (Taori et al., 2023) is a hybrid question-answer (QA) dataset containing 52k samples used for fine-tuning the **Alpaca** model. The dataset is automatically generated by the **self-instruct** (Wang et al., 2022) framework, which iteratively prompts the language model to generate new training instances given a few manually written instructions.

- **GSM8K** (Cobbe et al., 2021) evaluates the model's ability for multi-step mathematical reasoning with 8.5k linguistically diverse grade school math word problems.

- **SQUAD** (Rajpurkar, 2016) is a crowdsourced reading comprehension dataset based on Wiki articles. It contains over 100k QA pairs connected to over 500 articles.

- **Multi-News** (Fabbri et al., 2019) is a benchmark on multi-document summarization. It consists of 56k news articles and summary pairs where the news articles are extracted from newser.com and the summary is written by professional editors.

Furthermore, we introduced 10 LLMs of varying sizes into our problem, with their statistics shown in Table 3. All of the LLMs and their token costs here are accessed through the Together API [1].

### 3.2 DATA PREPROCESSING AND SPLITTING

Given the above dataset and LLMs, we construct a multi-task interaction dataset described in Sec 2.1. Specifically, we combine all the datasets of four tasks together first. For each query, we utilize ten LLMs in Sec 3.1 to answer it and obtain the corresponding response.

Table 2: Overview of Datasets.

| Dataset | Task Type | Metric | Cases |
|---|---|---|---|
| Alpaca | Hybrid QA | F1 | 600 |
| GSM8K | Reasoning | Accuracy | 600 |
| SQUAD | Reading Comprehension | F1 | 600 |
| Multi-News | Summary | F1 | 600 |

Then the response is compared with its ground truth to get its performance using the metric of each task described in Table 2. Furthermore, the cost is calculated with the number of input tokens and output tokens and the cost of different LLMs in Table 3. Here we utilize GPT-2 as in (Chen et al., 2023) to calculate the number of tokens.

After obtaining the multi-task interaction dataset, we split the dataset according to different experimental settings. We mainly have two major settings. The first is the standard setting, where all LLMs are visible in both the training and test sets, with some new queries appearing in the test set. The data is divided into training, validation, and test sets in a ratio of 70% : 10%: 20%, based on different queries. In the case of new LLM setting, we assume that the first six LLMs in Table 3 are observable, while the remaining four are new LLMs. Therefore, based on the standard setting, we first remove data related to the latter four LLMs from the training and validation sets, while keeping the test set as it is in the standard setting. Furthermore, following (Cao et al., 2023; Fey et al., 2023), we construct an auxiliary dataset with the interaction data of the four new LLMs on 80 queries sampled uniformly from the training set. This auxiliary dataset is not involved in the training process but serves as a few-shots during the testing phase.

---

[1] https://docs.together.ai/docs/inference-models

## 3.3 BASELINE METHODS

We compare our GRAPHROUTER model with the following baselines. We first compare `GraphRouter` with two rule-based baselines:

- **Largest LLM** always selects the largest LLM available.
- **Smallest LLM** always selects the smallest LLM available.

Then we compare `GraphRouter` with a prompt-based baseline:

- **Prompt LLM** incorporates the query, candidate models, and objectives (e.g., prioritizing effectiveness) directly into the prompt, and feeds it into an external LLM (e.g., GPT-4) to select the most suitable LLM from a pool of candidates.

Further, `GraphRouter` is compared with three representative routers for LLM selection:

- **Hybrid LLM** (Ding et al., 2024), when given a small LLM and a large LLM, trains a pre-trained language model to assign queries to the small or large model. We use LLaMA-2 (7b) and Llama-3.1-Turbo (70b) as our small and large LLM respectively, as they are the smallest and the largest LLM available. We also replace DeBERTa (He et al., 2020) with RoBERTa (Liu, 2019) as the pre-trained language model and observe better performance.
- **FrugalGPT** (Chen et al., 2023) utilizes a pre-trained language model to predict the score of the generation result of all LLMs given a query, and then selects the LLM with the highest score within a given cost. We also use RoBERTa (Liu, 2019) as the router's backbone model.
- **C2MAB-V** (Dai et al., 2024) uses a bandit-based model for LLM selection, which regards each LLM as an arm and implements an exploration mechanism to search for a better solution.

Finally, we set up a gold baseline as the optimal solution for LLM selection. The purpose of setting up the baseline is to see how far `GraphRouter` is from the optimal solution.

- **Oracle** defines the theoretical upper bound of the reward, where each query has been routed to the optimal LLM via oracle information.

## 3.4 METRICS

We utilize three metrics to evaluate the performance of `GraphRouter` and baselines.

- **Performance** is to evaluate the average quality of the responses across different queries given by each method, which is introduced in Sec 3.2 and Table 2.
- **Cost** evaluates the average LLM inference cost when responding to the queries, which is described in Sec 3.2.
- **Reward** is used to measure how well a method balances performance and cost. Different users may have varying levels of emphasis on performance and cost. Therefore, we define three scenarios: **Performance First, Balance, and Cost First**, to correspond to situations where users prioritize high performance, value both high performance and low cost equally, or prioritize low cost, respectively. Specifically, to eliminate the influence of scale, we first normalize both performance and cost. Then, we define the score as $Reward = \alpha \cdot Performance - \beta \cdot Cost$. For the three scenarios, we set the values of $\alpha$ and $\beta$ to (1, 0), (0.5, 0.5), and (0.2, 0.8), respectively.

Table 3: Statistics of different LLMs and their costs on Together API.

| LLM | Size | Cost per 1M tokens |
| --- | --- | --- |
| LLaMA-3 (7b) | 7b | 0.2 |
| Mixtral-8x7B | 56b | 0.6 |
| NousResearch | 34b | 0.8 |
| LLaMA-2 (7b) | 7b | 0.2 |
| Mistral-7b | 7b | 0.2 |
| LLaMA-3 (70b) | 70b | 0.9 |
| LLaMA-3-Turbo (8b) | 8b | 0.2 |
| LLaMA-3-Turbo (70b) | 70b | 0.9 |
| Llama-3.1-Turbo (70b) | 70b | 0.9 |
| Qwen-1.5 (72b) | 72b | 0.9 |

## 3.5 IMPLEMENTATION DETAILS

In the training stage, we set the graph neural network as a two-layer graph attention network, with a 32-dim hidden dimension. The batch size is 32, and the max training epoch is set to 1000. We use Adam optimizer (Diederik, 2014) for model training and gradually decay the learning rate from 1e-3

Table 4: **Comparison of Various Methods on Multi-task Interaction Dataset across Three Distinct Performance-Cost Weight Scenarios**. Bold and underline denote the best and second-best results. All datasets are evaluated on Performance, Cost, and Reward. Each number is the average of multiple rounds.

| Scenario | Performance First | | | Balance | | | Cost First | | |
|---|---|---|---|---|---|---|---|---|---|
| | Performance | Cost | Reward | Performance | Cost | Reward | Performance | Cost | Reward |
| Largest LLM | 0.431 | 0.871 | 0.431 | 0.431 | 0.871 | -0.220 | 0.431 | 0.871 | -0.701 |
| Smallest LLM | 0.279 | 0.031 | 0.279 | 0.279 | 0.031 | 0.124 | 0.279 | 0.031 | -0.009 |
| Prompt LLM | 0.474 | 0.812 | 0.474 | 0.285 | 0.0551 | 0.115 | 0.283 | 0.108 | -0.03 |
| Hybrid LLM | 0.510 | 0.871 | 0.510 | 0.470 | 0.451 | 0.009 | 0.276 | 0.151 | -0.066 |
| FrugalGPT | 0.517 | 0.671 | 0.517 | 0.400 | 0.072 | 0.164 | 0.411 | 0.031 | 0.057 |
| C2MAB-V | 0.479 | 0.871 | 0.479 | 0.423 | 0.031 | 0.196 | 0.279 | 0.031 | 0.031 |
| **GraphRouter** | 0.539 | 0.725 | **0.539** | 0.448 | 0.031 | **0.209** | 0.446 | 0.031 | **0.064** |
| Oracle | 0.588 | 0.586 | 0.588 | 0.504 | 0.040 | 0.231 | 0.483 | 0.031 | 0.072 |

to 0 with LambdaLR scheduler. We implement our proposed method using PyTorch[2] and PyG[3], and all the experiments are conducted on a single NVIDIA A100 Tensor Core GPU. As for LLMs, we rely on API calling from Together AI[4] to obtain responses.

# 4 EXPERIMENTAL RESULTS

## 4.1 COMPARISON WITH EXISTING BASELINES.

We compare `GraphRouter` with seven baselines in three scenarios in Table 4. We can observe that `GraphRouter` consistently and substantially surpasses existing routers, delivering a minimum effect improvement of 12.28% on metric Reward compared to the strongest baselines. Additionally, we observe that `GraphRouter` achieves at least 88.89% of the optimal solution (Table 4, row Oracle), further demonstrating the superiority of our framework. On the other hand, compared with the two rule-based LLM, `GraphRouter` achieves a better trade-off between Performance and Cost, therefore achieving a higher effect on Reward. Analyzing the effect of Prompt LLM, Hybrid LLM (Ding et al., 2024), and FrugalGPT (Chen et al., 2023), we demonstrate that without sufficient contextualized information, even LLM and trained moderate-size LM struggle to understand the query and candidates LLM effectively, even if we ignore their high inference costs. These results validate that effective usage of contextual information is crucial for selecting the optimal LLM.

## 4.2 GENERALIZATION ABILITY TO NEW LLMS

To validate the generalization ability of `GraphRouter` when facing new LLMs, we conduct experiments as described in Sec 3.2 in scenario **Balance**. To compare with other baselines, we add the auxiliary dataset into their training dataset. Specifically, we compare `GraphRouter` (few-shots) with HybridLLM, FrugalGPT, C2MAB-V, and `GraphRouter` (trained) on Reward and time cost (training time + inference time). We report our results in Table 5. We can observe that in comparison to the most costly **C2MAB-V (Dai et al., 2024)**, `GraphRouter` (few-shots) not only achieves substantial performance improvements in Reward by almost 10% but also greatly reduces Time Cost by over 99%. The amount of Reward Improvement and Time Cost Reduction has significantly surpassed those of other baselines. Additionally, compared to `GraphRouter` (trained), our approach significantly reduces time cost with only a slight performance loss. These observations demonstrate that `GraphRouter` is both effective and efficient in generalizing to new LLMs.

## 4.3 ABLATION STUDIES

**How does `GraphRouter` perform with varying sizes of GNN?** The size of a GNN is an important factor to consider when designing GNN algorithms. It not only affects the performance of the GNN but also introduces additional computational overhead if the size is too large. To find an optimal GNN size for `GraphRouter`, we explored sizes ranging from 16 to 80, as shown in Figure 6. As depicted

---

[2] https://pytorch.org/

[3] https://pytorch-geometric.readthedocs.io/en/latest/

[4] https://www.together.ai/

Table 5: **Comparison of methods in the few-shot setting on Reward, Time Cost, and the corresponding percentage Reward improvements and Time Cost reduction rate, relative to the most costly method (C2MAB-V (Dai et al., 2024)).**

| Method | Reward | Reward Improvement(%) | Time Cost | Time Cost Reduction(%) |
|---|---|---|---|---|
| HybridLLM | 0.01 | -94.71 | 273.45 | 49.57 |
| FrugalGPT | 0.171 | -9.52 | 63.15 | 88.35 |
| C2MAB-V | 0.189 | 0.00 | 542.25 | 0.00 |
| GraphRouter (few-shots) | 0.207 | 9.52 | 3.00 | 99.45 |
| GraphRouter (Trained) | 0.219 | 15.87 | 30.00 | 94.47 |

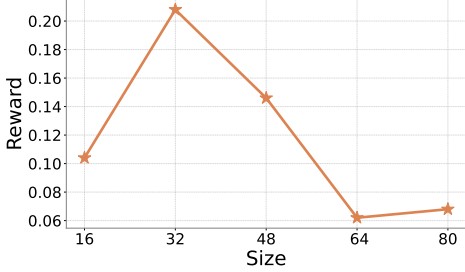

Figure 6: **The Reward of `GraphRouter` varies with the input size of the GNN.** Here, we set our in-channels equal to out-channels. As observed in the figure, the Reward of `GraphRouter` initially increases and then decreases, achieving the highest Reward when the size is 32.

Figure 7: **The Reward of `GraphRouter` varies with the number of GNN layers.** As shown in the figure, the Reward of `GraphRouter` initially increases and then decreases, reaching its highest Reward when the number of layers is 2.

in the figure, the Reward of `GraphRouter` initially improves as the size increases, reaching its peak at a size of 32, after which it starts to decline.

**What is the impact of different numbers of GNN layers on `GraphRouter`'s effectiveness?** The number of GNN layers has a significant impact on the expressiveness of the GNN. A shallow GNN struggles to learn deep contextual information, whereas an overly deep GNN can lead to issues such as over smoothing and overfitting. Moreover, increasing the number of layers also raises the computational cost. To determine the optimal number of GNN layers for `GraphRouter`, we conduct an exploration with the number of layers ranging from 0 to 5, as shown in Figure 7. As depicted in the figure, the Reward of `GraphRouter` initially improves with more layers but then declines, achieving its highest Reward when the number of layers is 2.

## 5 ADDITIONAL RELATED WORKS

**LLM selection.** With the scaling of the number of parameters in LLMs, their inference cost is also rapidly increasing. To optimize inference cost, several works have proposed using model switching to mitigate this issue. When given a small and a large LLM, Zhu et al. (2023) fine-tune a pre-trained language model as the model router to predict whether using a small LLM is sufficient. HybridLLM (Ding et al., 2024) improves on this by transforming training data to compensate for the influence of imbalanced labels. Other approaches move beyond choosing between two LLMs and introduce settings where multiple LLMs are present. Chen et al. (2023) train the router to predict the reliability score given a query and an LLM index. Šakota et al. (2024) generalize the router's training objective to accommodate the scenario where additional cost or performance constraints exist. Stripelis et al. (2024) examine the effectiveness of lighter routers built upon the k-nearest neighbors algorithm or Multilayer Perceptrons. C2MAB-V (Dai et al., 2024) employs a bandit-based model for LLM selection, treating each LLM as an arm and incorporating an exploration mechanism to find an improved solution. Different from previous approaches, where the router only learns from query-model interaction, our GRAPHROUTER fully utilizes the information in the training data by

jointly modeling the query-model, query-query, and model-model relationship. This allows us to learn effective representations for tasks, queries, and models, enabling better generalizability.

**Graph for modeling relationships.** Graphs have demonstrated great potential in modeling complex relationships (Fey et al., 2023; Cao et al., 2023; Gao & Xu, 2020; Chen et al., 2022; Wu et al., 2022b; Yang et al., 2021). Solving relational data with graphs often involves extracting nodes and edges from the data, and then modeling their relationships with embeddings. Traditional graph algorithms, such as label propagation (Xie et al., 2022; Zhu & Ghahramani, 2002), directly utilize edge relationships to propagate known labels to target nodes. With the advancement of deep learning, graph neural networks (GNNs) have become the more popular approach for researchers to model relationships within data. They are also found to have widespread application in fields such as recommendation systems (Min et al., 2022) and social networks (Wu et al., 2020). GNNs can be broadly classified into Message Passing Neural Networks (MPNNs), which include models like GCN (Kipf & Welling, 2017), GraphSAGE (Hamilton et al., 2017), and GAT (Veličković et al., 2017), as well as non-MPNN architectures (Wu et al., 2022a; Ying et al., 2021). Additionally, Heterogeneous Graph Neural Networks (HeterGNNs) (Peng et al., 2019; Hu et al., 2020; Schlichtkrull et al., 2017) and Heterogeneous Graph Attention Networks (HGATs) (Wang et al., 2019) have also been proposed to handle more complex graph data. In recent years, researchers have begun exploring the zero-shot or few-shot capabilities of GNNs (Fey et al., 2023; Cao et al., 2023; Gao & Xu, 2020; Chen et al., 2022), aiming to address more complex real-world challenges, such as the cold start problem in recommendation systems. Building on these studies, we incorporate GNNs' powerful ability to represent contextual heterogeneous relationships and their zero-shot capabilities into the LLM selection problem.

## 6 CONCLUSION AND DISCUSSION

**(1) Conclusion.** We present `GraphRouter`, a graph inductive framework for LLM routing during inference with multiple LLMs. This work is the first to address the LLM routing problem by reframing it as an edge prediction task between query nodes and LLM nodes. Using graph structure, we fully capture contextual information from prior interaction data to learn effective task, query, and graph representations. Through our experiments on the combined dataset from four open-domain QA datasets, and with three different application scenarios, we demonstrated the superiority of `GraphRouter` over competitive LLM selection baselines and showed that our framework is on par with the ideal "God's-eye view" solution. Beyond traditional settings, we also tested our framework in a more challenging setting where new LLMs were introduced during test time, and we demonstrated `GraphRouter`'s strong generalization ability compared to previous baselines. We hope that `GraphRouter`, along with our approach of incorporating interaction data through graphs, will facilitate future research on LLM routing. **(2) Limitations.** This work primarily serves as exploratory work to validate the idea of how modeling past interaction data in a graph could enhance the process of LLM selection. We acknowledge that leveraging more complex graph signals, such as paths, or the taxonomy of LLMs (e.g., family trees like LLaMA2 → LLaMA3 → LLaMA3.1 (Touvron et al., 2023)), could further improve `GraphRouter`'s performance, but that is beyond the scope of this paper, and we leave it for future work. **(3) Future Work.** Some other interesting questions to explore in the future include: 1) When answering complex queries, prompting methods like **Chain-of-Thought** (Wei et al., 2022), **Tree-of-Thought** (Yao et al., 2024) are also widely used to enhance the reasoning ability of LLMs. Given the vast number of these methods, predicting the generation result and selecting the best one remains a great challenge. On the other hand, selecting the prompting method is similar to selecting the best LLM for inference, as we are both aiming to predict the generation result based on past interaction data. As a result, it is interesting to explore whether `GraphRouter` could also be adapted to this task. 2) In a multi-agent system, it is critical to choose the appropriate LLM for each module on a specific query and task. It would be valuable to conduct experiments on whether an inductive graph framework like `GraphRouter` could also excel on this challenging task, where multiple LLMs are being selected simultaneously. 3) How to enable LLMs to better understand numerical differences is a direction for future consideration in modeling LLM features more effectively. Using current LLMs to model and understand numerical information like token pricing and context length (Romera-Paredes et al., 2024; Ahn et al., 2024; Imani et al., 2023; Lewkowycz et al., 2022) is still an open question. These numerical details are significant factors affecting the performance of LLM selection.

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

## A  DESCRIPTION FOR TASKS AND LLMS

Using descriptions generated by LLM and embeddings derived from BERT as initial node embeddings for GNNs enhances their expressiveness and generalization capabilities. Here, we have listed descriptions of different tasks and various LLMs obtained using GPT-4o. Specifically, GPT-4o provides insights into the unique characteristics and challenges of different tasks, as well as the size, cost, and particular strengths of different LLMs. The detailed descriptions are shown in the table below.

Table 6: **Description of Alpaca task**.

The Alpaca dataset is designed for instruction-following tasks, where the model is required to generate coherent and contextually appropriate responses to given instructions or prompts. It focuses on understanding diverse user requests and providing informative and accurate outputs based on those instructions.

Table 7: **Description of GSM8K task**.

The GSM8K dataset is tailored for mathematical problem-solving tasks. It consists of natural language math problems that require the model to comprehend the problem statement, apply the correct mathematical operations, and provide the solution. The primary challenge lies in both parsing complex language and performing accurate calculations.

Table 8: **Description of SQUAD task**.

The SQuAD dataset is focused on question-answering tasks, where the model is given a passage of text and needs to extract or generate a precise answer to a question based on the content of the passage. The dataset emphasizes comprehension, retrieval of relevant information, and concise answer generation.

## B  EXPERIMENTS ON DIFFERENT DATASETS AND API SETTINGS

### B.1  GENERALIZATION CAPABILITY ON LARGER DATASETS

To validate whether our method is applicable to additional tasks and LLM models, we expanded upon the dataset from Section 3.1 by adding two new datasets. The first, HumanEval (Chen et al., 2021), is a dataset that measures LLMs' coding capabilities, specifically consisting of 164 original programming problems that assess language comprehension, algorithms, and simple mathematics, with some problems comparable to basic software interview questions. The second dataset is HotpotQA (Yang et al., 2018), a question answering dataset with 113k entries featuring natural, multi-hop questions with strong supervision for supporting facts to enable more explainable question answering systems. Similarly, we summarized the metrics and data volume for these datasets under our experimental settings, as shown in Table 20. Additionally, we incorporated four more LLM models from the Together AI API, based on the LLM models in Section 3.1: Qwen-2 (72b), Code Llama (34b), Mixtral-8x22B, and Upstage. Likewise, the size and cost information of these LLMs are summarized in Table 21. Further, similar to section 3.2, we constructed a dataset based on the extended dataset and the interaction data from LLMs, which was then divided into training, validation, and test sets in a ratio of 70% : 10% : 20%, based on different queries. We compared the performance of `GraphRouter` with other baselines on this extended dataset and reported the experimental results in Table 22. We observed that `GraphRouter` improved the Reward by at least 12.3% compared to

Table 9: **Description of Multi-News task**.

The Multi-News dataset is aimed at text summarization tasks. It contains multiple news articles on the same topic, and the model's objective is to generate a concise and comprehensive summary that integrates information from all the articles. The challenge is to distill key points while maintaining coherence and avoiding redundancy.

Table 10: **Description of LLaMA-3 (7b)**.

This is a relatively small-sized model (7 billion parameters) designed for general-purpose language tasks. Its low cost per million tokens (0.2) makes it an affordable option for many applications requiring quick responses with moderate accuracy.

the baselines, confirming that `GraphRouter`'s approach can be generalized to more datasets and other LLMs.

## B.2    DISCRIMINATIVE ABILITY FOR SIMILAR LLMS

To explore whether `GraphRouter` can effectively model and differentiate between similar LLMs, we extracted data related to the LLaMA series of LLMs from the interaction dataset introduced in section 3.2 for training and prediction. Specifically, we extracted interaction data for LLMs including LLaMA-3 (7b), LLaMA-2 (7b), LLaMA-3 (70b), LLaMA-3-Turbo (8b), LLaMA-3-Turbo (70b), and Llama-3.1-Turbo (70b). We compared the performance of `GraphRouter` and the best-performing baseline, FrugalGPT, on this dataset and reported the specific results in Table 23. We observed that, compared to FrugalGPT, `GraphRouter` improved the Reward by at least 10.8%. These observations demonstrate that `GraphRouter` can effectively capture differences in the capabilities of different LLMs through interactions, achieving good results.

Table 11: **Description of Mixtral-8x7B**.

With a combined size of 56 billion parameters, this model aims to provide stronger language modeling capabilities. Its cost per million tokens is 0.6, reflecting its balance between performance and affordability for more complex tasks.

Table 12: **Description of NousResearch (34b)**.

A mid-sized model with 34 billion parameters, suitable for handling moderately complex language tasks. Its cost is higher at 0.8 per million tokens, indicating a greater computational demand, likely due to its enhanced capabilities over smaller models.

Table 13: **Description of LLaMA-2 (7b)**.

A compact model at 7 billion parameters, it offers similar capabilities and pricing to LLaMA-3 (7b) at a cost of 0.2 per million tokens. It's an efficient choice for tasks requiring decent performance without high computational costs.

Table 14: **Description of Mistral-7b**.

With 7 billion parameters, Mistral-7b is optimized for lightweight tasks, balancing speed and efficiency. Its cost per million tokens is 0.2, making it cost-effective for standard use cases without the need for complex computations.

Table 15: **Description of LLaMA-3 (70b)**.

A larger variant of LLaMA-3, this model has 70 billion parameters, providing advanced capabilities for complex tasks. Its cost per million tokens is 0.9, indicating its higher computational demand and enhanced performance.

Table 16: **Description of LLaMA-3-Turbo (8b)**.

A variant optimized for speed and efficiency with 8 billion parameters. Its cost per million tokens is only 0.2, suggesting that it is designed to handle tasks quickly while being highly cost-effective.

Table 17: **Description of LLaMA-3-Turbo (70b)**.

This model, at 70 billion parameters, is tailored for high performance with an emphasis on efficiency. The cost is 0.9 per million tokens, reflecting its advanced capabilities for a broad range of tasks requiring more computation.

Table 18: **Description of Llama-3.1-Turbo (70b)**.

Large model with 70 billion parameters, likely to offer strong capabilities for various language tasks. Its cost is also 0.9 per million tokens, suggesting similar performance and computational needs as other 70b models.

Table 19: **Description of Qwen-1.5 (72b)**.

With 72 billion parameters, Qwen-1.5 is among the largest models in the list, designed for high-complexity tasks. Its cost per million tokens is 0.9, making it comparable to other high-performance models in terms of both capability and expense.

Table 20: Overview of extended datasets.

| Dataset | Task Type | Metric | Cases |
|---|---|---|---|
| Alpaca | Hybrid QA | F1 | 600 |
| GSM8K | Reasoning | Accuracy | 600 |
| SQUAD | Reading Comprehension | F1 | 600 |
| Multi-News | Summary | F1 | 600 |
| HumanEval | Code | Pass@1 | 600 |
| HotpotQA | Multi-hop QA | EM | 600 |

Table 21: Statistics of larger LLMs set and their costs on Together API.

| LLM | Size | Cost per 1M tokens |
|---|---|---|
| LLaMA-3 (7b) | 7b | 0.2 |
| Mixtral-8x7B | 56b | 0.6 |
| NousResearch | 34b | 0.8 |
| LLaMA-2 (7b) | 7b | 0.2 |
| Mistral-7b | 7b | 0.2 |
| LLaMA-3 (70b) | 70b | 0.9 |
| LLaMA-3-Turbo (8b) | 8b | 0.2 |
| LLaMA-3-Turbo (70b) | 70b | 0.9 |
| Llama-3.1-Turbo (70b) | 70b | 0.9 |
| Qwen-1.5 (72b) | 72b | 0.9 |
| Qwen-2 (72b) | 72b | 0.9 |
| Code Llama (34b) | 34b | 0.8 |
| Mixtral-8x22B | 176b | 1.2 |
| Upstage | 11b | 0.3 |

Table 22: **Comparison of various methods on large multi-task interaction dataset across Three Distinct Performance-Cost Weight Scenarios.** . Bold denotes the best results. Each metric reflects average values from multiple evaluation rounds.

| Model | Performance First | | | Balance | | | Cost First | | |
|---|---|---|---|---|---|---|---|---|---|
| | Performance | Cost | Reward | Performance | Cost | Reward | Performance | Cost | Reward |
| Largest LLM | 0.321 | 0.611 | 0.321 | 0.321 | 0.611 | -0.145 | 0.321 | 0.611 | -0.425 |
| Smallest LLM | 0.180 | 0.018 | 0.180 | 0.180 | 0.018 | 0.081 | 0.180 | 0.018 | 0.022 |
| Prompt LLM | 0.260 | 0.654 | 0.260 | 0.180 | 0.026 | 0.077 | 0.184 | 0.024 | 0.018 |
| Hybrid LLM | 0.350 | 0.611 | 0.350 | 0.311 | 0.256 | 0.028 | 0.184 | 0.103 | -0.046 |
| FrugalGPT | 0.277 | 0.357 | 0.277 | 0.259 | 0.143 | 0.058 | 0.272 | 0.034 | 0.027 |
| C2MAB-V | 0.254 | 0.366 | 0.254 | 0.295 | 0.122 | 0.087 | 0.261 | 0.036 | 0.023 |
| **GraphRouter** | 0.393 | 0.220 | **0.393** | 0.297 | 0.052 | **0.122** | **0.299** | 0.018 | **0.046** |
| Oracle | 0.432 | 0.429 | 0.432 | 0.432 | 0.398 | 0.171 | 0.330 | 0.018 | 0.052 |

Table 23: **Comparison of various methods on LLaMA-series dataset.** Bold denotes the best results. Each metric reflects average values from multiple evaluation rounds.

| Model | Performance First | | | Balance | | | Cost First | | |
|---|---|---|---|---|---|---|---|---|---|
| | Performance | Cost | Reward | Performance | Cost | Reward | Performance | Cost | Reward |
| FrugalGPT | 0.382 | 0.299 | 0.382 | 0.367 | 0.043 | 0.162 | 0.372 | 0.030 | 0.050 |
| **GraphRouter** | 0.422 | 0.307 | **0.422** | 0.416 | 0.032 | **0.192** | 0.416 | 0.032 | **0.058** |
| Oracle | 0.489 | 0.376 | 0.489 | 0.459 | 0.051 | 0.204 | 0.436 | 0.032 | 0.062 |

