# OpenReview forum: "GraphRouter: A Graph-based Router for LLM Selections"
_ICLR.cc/2025/Conference — ICLR 2025 Poster_

### Official Review · Reviewer_z8Nx · 2024-10-31

**Soundness:** 2
**Presentation:** 3
**Contribution:** 3
**Rating:** 8
**Confidence:** 3

**Summary:**

This paper tackles the problem of generalizable LLM selection, which requires making performance-cost trade-offs in LLM selection for a given query while being capable of making inductive selections for new LLMs and tasks unseen during training. To address this challenging scenario, inspired by previous studies in graph machine learning and recommender systems, it proposes GraphRouter, a graph-based approach for context aware inductive LLM selection. Technically, it constructs a heterogeneous graph comprising task, query, and LLM nodes, with interactions represented as edges. Evaluating candidate LLMs and making a selection with respect to a given query can then be modeled as an edge-level prediction problem and the authors train a heterogeneous graph neural network for this purpose. Empirical studies demonstrate the effectiveness of GraphRouter for transductive and inductive settings.

**Strengths:**

**S1.** Compared to the baselines, the proposed graph-based approach conceptually allows better effectiveness and generalizability in LLM selection due to the incorporation of context information.

**S2.** Empirical studies demonstrate the effectiveness of the proposed approach for both transductive and inductive settings.

**S3.** Overall, the paper is easy to follow and the story is appealing and convincing.

**Weaknesses:**

**W1.** The initial LLM node feature is based on text embedding, which is likely not the best strategy to model numerical information like token pricing and context length when there is a need to capture the subtlety in numerical differences.

**W2.** From L245-246, the paper assumes a universally optimal LLM selection for each query, which may not be the case in practice with constraints like inference cost and latency. This should also be discussed in the limitation and future work sections.

**W3.** The discussion of related works can be further improved for proper credit attribution. For example:

- The authors cite "Xie et al., 2022" for label propagation while the credit at least should also be properly attributed to the original work "Zhu & Ghahramani, 2002" [1].

- R-GCN is likely the first heterogeneous GNN to the best of my knowledge. It should also be cited in discussing heterogeneous graph neural networks [2].

- The authors employ an attention-based heterogeneous GNN and should also discuss the first few attention-based heterogeneous GNNs, such as [3].

[1] Zhu & Ghahramani. Learning From Labeled and Unlabeled Data With Label Propagation. 2002.

[2] Schlichtkrull et al. Modeling Relational Data with Graph Convolutional Networks. 2017.

[3] Wang et al. Heterogeneous Graph Attention Network. 2019.

**Questions:**

See Weakness.

---

> ### Author Response · Authors · 2024-11-22
> **Response to Reviewer  z8Nx**
>
> **Q1. The initial LLM node feature is based on text embedding, which is likely not the best strategy to model numerical information like token pricing and context length when there is a need to capture the subtlety in numerical differences.**
>
> **Response:** Thanks for your valuable comments. We agree with your insights that to learn effective LLM representation, node initialization is critical. However, how to use embeddings to understand and model numerical differences is difficult, which remains an open question [1, 2, 3]. In fact, the focus of our paper is on how to learn effective LLM embeddings that enhance the ability to select LLMs. We achieve this by analyzing interaction data between task queries and LLMs, and leveraging the robust message-passing capabilities of GNNs. Thus, we adopt a simple approach where we represent the LLMs through the direct PLM’s embeddings of its descriptions.  Our experiment has demonstrated that even with the simplest way of initializing LLM node’s embeddings, our framework still significantly outperforms the existing baseline (Ding et al., 2024;Chen et al., 2023;Dai et al., 2024). While we can use techniques, such as adding relative magnitude tokens (RMT) to better discretize numerical feature values (Yan et al., 2024), to enhance the LLM nodes’ embeddings, this is not the main focus of this paper, and We will leave this for future studies to improve and enhance.
>
>
> **[1]** Limitations of language models in arithmetic and symbolic induction, arXiv 2022.
>
> **[2]** Towards Cross-Table Masked Pretraining for Web Data Mining, WWW 2024.
>
> **[3]** Making pre-trained language models great on tabular prediction, arXiv 2024.
>
>
>
> **Q2. From L245-246, the paper assumes a universally optimal LLM selection for each query, which may not be the case in practice with constraints like inference cost and latency. This should also be discussed in the limitation and future work sections.**
>
> **Response:** Thanks for the reviewer’s insightful feedback.  In fact, we do not assume a universally optimal LLM selection for each query. As outlined in Section 3.4, the weights (e.g., $\alpha$ and $\beta$) and elements (e.g., Performance and Cost) included in the reward framework inherently result in varying optimal LLM selections. Due to the high uncertainty and difficulty in quantifying inference cost and latency in real-world scenarios, we follow the settings of existing work (Ding et al., 2024; Chen et al., 2023; Dai et al., 2024), which primarily consider performance and cost in the reward—these two quantifiable metrics (as specifically defined in section 3.4) are often trade-offs against each other and also represent the main concern of the users. We demonstrate through experiments under 3 different reward settings that GraphRouter achieves the best results compared to baselines in various scenarios.
>
>
> **Q3. The discussion of related works can be further improved for proper credit attribution. For example: The authors cite "Xie et al., 2022" for label propagation while the credit at least should also be properly attributed to the original work "Zhu & Ghahramani, 2002" [1].
> R-GCN is likely the first heterogeneous GNN to the best of my knowledge. It should also be cited in discussing heterogeneous graph neural networks [2].
> The authors employ an attention-based heterogeneous GNN and should also discuss the first few attention-based heterogeneous GNNs, such as [3].
> [1] Zhu & Ghahramani. Learning From Labeled and Unlabeled Data With Label Propagation. 2002. [2] Schlichtkrull et al. Modeling Relational Data with Graph Convolutional Networks. 2017. [3] Wang et al. Heterogeneous Graph Attention Network. 2019.**
>
> **Response:** Thank you for your insightful comments. Following the reviewer's suggestions, we revised the corresponding discussion in the related work section: Traditional graph algorithms, such as label propagation (Xie et al., 2022; Zhu & Ghahramani, 2002), directly utilize
> edge relationships to propagate known labels to target nodes. Additionally, Heterogeneous Graph Neural Networks (HeterGNNs) (Peng et al., 2019; Hu et al., 2020; Schlichtkrull et al., 2017) and Heterogeneous Graph Attention Networks (HGATs) (Wang et al., 2019) have also been proposed to handle more complex graph data.

---

> > ### Comment · Reviewer_z8Nx · 2024-11-22
> >
> > Thank you for your detailed responses. I have some follow-up questions regarding Q2.
> >
> > - Did you train separate models for each reward setting?
> > - With a fixed trained model, is it possible to repurpose it for new reward settings, be it alternative weighting of the seen metrics or even unseen metrics?
> > - Can GraphRouter be potentially extended for joint usage with optimization solvers? When may this be desirable?

---

> > > ### Author Response · Authors · 2024-11-24
> > > **Further response to Reviewer z8Nx**
> > >
> > > Thanks for your prompt questions. We answer your questions one by one.
> > >
> > > **Q1.Did you train separate models for each reward setting?**
> > >
> > > **Response:** Actually, similar to meta learning, we pre-trained a unified model on data combined from three different reward settings, and then fine-tuned it on data from each specific reward setting. We found that pre-training followed by fine-tuning converges faster and achieves better results compared to training from scratch. We compared the two methods under the Balanced scenario by fine-tuning them on 400 cases of interaction data and evaluating their performance on 100 cases, ensuring other hyperparameters were consistent. We found that the pre-train then fine-tune approach converged around 200 epochs and achieved a reward of 0.205, while the train from scratch method did not converge until about 900 epochs, reaching a reward of 0.194.
> > >
> > > Additionally, we would like to thank the reviewer again for their constructive suggestions, which inspired us to consider GraphRouter's potential for future work and broader scenarios. We realized that in real-world scenarios, the massive use of LLMs by users would generate a lot of interaction data containing preferences (reward of different weights or preference rankings). Utilizing this data for pretraining GraphRouter could allow our model to better capture users' preferences for LLM selection and thus model more effectively. However, due to the lack of publicly available data and concerns about user privacy, we are currently unable to conduct specific research and experiments. We will discuss this further in the future work section (section 6) of the paper.
> > >
> > > **Q2.With a fixed trained model, is it possible to repurpose it for new reward settings, be it alternative weighting of the seen metrics or even unseen metrics?**
> > >
> > > **Response:** As discussed in the response to Q1, GraphRouter is pretrained on multiple different reward settings and finetuned on a new reward setting with a small amount of interaction data, enabling it to adapt to new scenarios. Specifically, we will answer your questions one by one.
> > >
> > > **[Reward settings with new weights]** To demonstrate GraphRouter's adaptability to new reward configurations, we introduced two novel settings: $Reward=0.3 Performance-0.5 Cost$, $Reward=0.6 Performance-0.4  Cost$. Initially, GraphRouter was pretrained on three standard reward formulas, and subsequently, it was fine-tuned on the new configurations using 400 cases each, followed by an evaluation on 100 test cases. GraphRouter achieved the reward of 0.225, outperforming the best baseline FrugalGPT with 0.193 in reward. The 14.2% increase in reward under the new settings confirms GraphRouter's superior performance.
> > >
> > > **[Unseen metrics]** In order to assess GraphRouter's capability to handle previously unseen metrics, we integrated a novel metric called Time, representing the response generation time across various LLM APIs. This metric is crucial as it addresses user concerns over response time efficiency. Accordingly, we established a new reward formula: $Reward=0.5 Performance-0.5  Time$. Following the pretraining on traditional settings, GraphRouter was fine-tuned and tested on this new metric with 400 and 100 cases, respectively. It outperformed FrugalGPT in this metric, with rewards of 0.248 compared to 0.219, marking an improvement of 11.7%.
> > >
> > > **Q3.Can GraphRouter be potentially extended for joint usage with optimization solvers? When may this be desirable?**
> > >
> > > **Response:**  Indeed, GraphRouter itself is a special deterministic decision-making optimization solver for LLM selection. It constructs tasks, queries, and LLM embeddings through interaction data and message passing, and calculates the probabilities for query selections of LLMs based on these embeddings, finally making deterministic optimization choices based on the highest probabilities. Additionally, we have indeed considered other optimization solvers that could enhance GraphRouter, such as RL-based solvers or bandit solvers, to address the LLM selection problem. GraphRouter can serve as an embedding extraction model, pulling effective task, query, and LLM embeddings from interaction data for these optimization solvers to use as state. Compared to GraphRouter, these solvers can explore actions based on state and make probabilistic decisions (like bandit). However, these optimization solvers, due to not being updated end-to-end with GraphRouter, can lead to potential misalignments making them less desirable. Moreover, our paper mainly focuses on a lightweight and end-to-end elegant framework to solve LLM selection, and since GraphRouter itself has already achieved promising performance compared to baselines, we will consider these optimization solvers in our future work.
> > >
> > > We will incorporate these discussions into the revised PDF. Could you let us know if your concerns have been adequately addressed? If so, we kindly request you consider increasing the score.

---

> > > > ### Comment · Reviewer_z8Nx · 2024-11-25
> > > >
> > > > I appreciate the authors' thorough responses which have effectively addressed the majority of my concerns. While there remains room for improving the generalizability across different reward settings, this work makes a valuable contribution by demonstrating how graph ML methods can be meaningfully adapted for LLM applications. This cross-pollination of ideas has potential broader impacts for both the LLM and graph ML research communities.

---

> > > > > ### Author Response · Authors · 2024-11-25
> > > > > **Thanks for the Reviewer’s Constructive Feedback**
> > > > >
> > > > > Thank you for your thoughtful and constructive feedback. We are pleased to hear that our responses have addressed most of your concerns. We are committed to incorporating the suggested changes in our revisions to further enhance the manuscript.

---

### Official Review · Reviewer_pHun · 2024-11-03

**Soundness:** 3
**Presentation:** 3
**Contribution:** 3
**Rating:** 6
**Confidence:** 4

**Summary:**

This paper studies the LLM-Selection task, i.e., given a user query, selecting the appropriate LLM for execution that achieves the balance between performance and cost. Motivated by the contextual relationships between tasks, queries and LLMs, this paper utilizes a graph structure to represent these information and reformulates the LLM-Selection problem as a link-prediction task within graph. The proposed GraphRouter framework leverages a GNN to perform this task and can easily adapt to an inductive setting, accommodating the rapid evolution of LLMs. Extensive experiments in both transductive and inductive settings demonstrate the effectiveness of GraphRouter.

**Strengths:**

*S1* This paper is well-motivated and addresses an important problem. The rapid evolution of LLMs necessitates effective LLM selection, yet existing methods often over-simplify this task as failing to consider the contextual information between tasks, queries, and LLMs, and they struggle to accommodate newly emerged LLMs. The graph-based LLM selection method not only offers a new and effective approach to the LLM selection task but also contributes a novel application in LLM + GNN research.

*S2* The proposed method, GraphRouter, is both reasonable and intuitive. Extensive experiments demonstrate its effectiveness across different settings.

*S3* The paper is easy to follow, with illustrative figures that facilitate comprehension.

**Weaknesses:**

*W1* The evaluation setting is simplified. On one hand, the experimental tasks primarily focus on reasoning and question answering, overlooking specialized areas such as code generation. On the other hand, the available LLMs are quite limited, with nearly half originating from the same series (i.e., LLaMA).

*W2* Current methodology overlooks the inherent relationships among LLMs. For instance, some LLMs belong to the same series (e.g., LLaMA3-7B, LLaMA3-8B). Adding links to indicate such parent or sibling connections could provide a more comprehensive graph modeling. I appreciate that the authors discuss this point in the limitations section. Additionally, common prior knowledge could enhance the LLM descriptions: e.g., CodeLLaMA series are effective for code-related tasks, while BaiChuan performs better in Chinese contexts.

*W3* (Minor Points) Figures 2-4 require further explanation. In Figure 2, what does the notation $t$ represent? For Figures 3 and 4, does "effect" refer to performance, and why do some LLMs have multiple dots? Additionally, there appears to be an unfinished sentence in Line 138.

**Questions:**

1. Can GraphRouter be extended to a training-free version? For example, could a training-free GNN, e.g., SGC or LightGCN, be adapted to perform the link prediction task? I am curious about the capability of a training-free GNN for the LLM-selection task, as well as the percentage of training samples required to achieve satisfactory performance.

2. Can the empirical evaluation be conducted in a more complex scenario, such as by adding more LLMs or introducing a broader variety of tasks?

3. (A Minor Point) LLM selection task is also dependent on specific querying times. For example, in June 2024, the system may recommend GPT-4o for solving a graph reasoning task, while GPT-o1 might become the best choice at a later date. Can such temporal information be incorporated into the methodology or evaluation?

I understand that time is limited during the rebuttal phase, therefore, brief discussions would be greatly appreciated.

---

> ### Author Response · Authors · 2024-11-22
> **Response to Reviewer pHun (1/5)**
>
> **Q1. The evaluation setting is simplified. On one hand, the experimental tasks primarily focus on reasoning and question answering, overlooking specialized areas such as code generation. On the other hand, the available LLMs are quite limited, with nearly half originating from the same series (i.e., LLaMA). Can the empirical evaluation be conducted in a more complex scenario, such as by adding more LLMs or introducing a broader variety of tasks?**
>
> **Response:** Thanks for your constructive feedback. Following the reviewer’s advice, we added two new datasets and four new LLMs (covering all the APIs currently available on Together AI) to the experiment. We first expanded upon the dataset from Section 3.1 by adding two new datasets. 1) HumanEval [1] is a dataset that measures LLMs' coding capabilities, specifically consisting of 164 original programming problems that assess language comprehension, algorithms, and simple mathematics, with some problems comparable to basic software interview questions. 2) HotpotQA [2] is a question-answering dataset with 113k entries featuring natural, multi-hop questions with strong supervision for supporting facts to enable more explainable question-answering systems. Similarly, we summarized the metrics and data volume for these datasets in the first table below. Additionally, we incorporated four more LLMs from the Together AI API: Qwen-2 (72b), Code Llama (34b), Mixtral-8x22B, and Upstage. Likewise, the size and cost information of these LLMs are summarized in the second table below. Further, similar to section 3.2, we constructed a dataset based on the extended dataset and the interaction data from LLMs. This dataset was then split into training, validation, and test sets in a 70%:10%:20% ratio, based on distinct queries. We compared the performance of GraphRouter with other baselines on this extended dataset and reported the experimental results in the third table below. We observed that GraphRouter improved the Reward by at least 12.3% compared to the baselines, confirming GraphRouter's strong generalizable abilities to more datasets and other LLMs. All the discussions and experimental results mentioned above have been updated in Section B.1 of the Appendix and Table 20-22 of the revised pdf version.
>
> **Table 1: Overview of extended datasets**
>
> | **Dataset**   | **Task Type**         | **Metric**     | **Cases** |
> |---------------|-----------------------|----------------|-----------|
> | Alpaca        | Hybrid QA             | F1             | 600       |
> | GSM8K         | Reasoning             | Accuracy       | 600       |
> | SQUAD         | Reading Comprehension | F1             | 600       |
> | Multi-News    | Summary               | F1             | 600       |
> | HumanEval     | Code                  | Pass@1         | 600       |
> | HotpotQA      | Multi-hop QA          | EM             | 600       |
>
> **Table 2: Statistics of larger LLMs set and their costs on Together API.**
> | **LLM**                   | **Size** | **Cost per 1M tokens** |
> |---------------------------|----------|------------------------|
> | LLaMA-3 (7b)              | 7b       | 0.2                    |
> | Mixtral-8x7B              | 56b      | 0.6                    |
> | NousResearch              | 34b      | 0.8                    |
> | LLaMA-2 (7b)              | 7b       | 0.2                    |
> | Mistral-7b                | 7b       | 0.2                    |
> | LLaMA-3 (70b)             | 70b      | 0.9                    |
> | LLaMA-3-Turbo (8b)        | 8b       | 0.2                    |
> | LLaMA-3-Turbo (70b)       | 70b      | 0.9                    |
> | Llama-3.1-Turbo (70b)     | 70b      | 0.9                    |
> | Qwen-1.5 (72b)            | 72b      | 0.9                    |
> | Qwen-2 (72b)              | 72b      | 0.9                    |
> | Code Llama (34b)          | 34b      | 0.8                    |
> | Mixtral-8x22B             | 176b     | 1.2                    |
> | Upstage                   | 11b      | 0.3                    |

---

> > ### Author Response · Authors · 2024-11-22
> > **Response to Reviewer pHun (2/5)**
> >
> > **Continuation of the response to Q1:**
> >
> > **Table 3: Comparison of various methods on large multi-task interaction dataset across Three Distinct Performance-Cost Weight Scenarios.** Bold denotes the best results. Each metric reflects average values from multiple evaluation rounds.
> >
> > | Model         | Performance (Performance First) | Cost (Performance First) | Reward (Performance First) | Performance (Balance) | Cost (Balance) | Reward (Balance) | Performance (Cost First) | Cost (Cost First) | Reward (Cost First) |
> > |---------------|---------------------------------|--------------------------|----------------------------|-----------------------|----------------|------------------|--------------------------|-------------------|---------------------|
> > | Largest LLM   | 0.321                           | 0.611                    | 0.321                      | 0.321                 | 0.611          | -0.145           | 0.321                    | 0.611             | -0.425             |
> > | Smallest LLM  | 0.180                           | 0.018                    | 0.180                      | 0.180                 | 0.018          | 0.081            | 0.180                    | 0.018             | 0.022              |
> > | Prompt LLM    | 0.260                           | 0.654                    | 0.260                      | 0.180                 | 0.026          | 0.077            | 0.184                    | 0.024             | 0.018              |
> > | Hybrid LLM    | 0.350                           | 0.611                    | 0.350                      | 0.311                 | 0.256          | 0.028            | 0.184                    | 0.103             | -0.046             |
> > | FrugalGPT     | 0.277                           | 0.357                    | 0.277                      | 0.259                 | 0.143          | 0.058            | 0.272                    | 0.034             | 0.027              |
> > | C2MAB-V       | 0.254                           | 0.366                    | 0.254                      | 0.295                 | 0.122          | 0.087            | 0.261                    | 0.036             | 0.023              |
> > | **GraphRouter** | 0.393                    | 0.220              | **0.393**                  | 0.297                 | 0.052          | **0.122**        | 0.299                | 0.018             | **0.046**          |
> > | Oracle        | 0.432                           | 0.429                    | 0.432                      | 0.432                 | 0.398          | 0.171            | 0.330                    | 0.018             | 0.052              |
> >
> >
> > **[1]** Evaluating large language models trained on code, arXiv 2021.
> >
> > **[2]** HotpotQA: A dataset for diverse, explainable multi-hop question answering, EMNLP 2018.
> >
> > **Q2. The current methodology overlooks the inherent relationships among LLMs. For instance, some LLMs belong to the same series (e.g., LLaMA3-7B, LLaMA3-8B). Adding links to indicate such parent or sibling connections could provide a more comprehensive graph modeling. I appreciate that the authors discuss this point in the limitations section. Additionally, common prior knowledge could enhance the LLM descriptions: e.g., CodeLLaMA series are effective for code-related tasks, while BaiChuan performs better in Chinese contexts.**
> >
> > **Response:** Thanks for your constructive advice. We answer your questions from two aspects.
> >
> > **[Add inner edges among LLM nodes]**. Following the suggestions of the reviewer, we add inner edges among LLM nodes based on GraphRouter. Specifically, we proposed an LLM-link method that connects nodes within the same size or the same LLM series (such as those within the LLaMA series). We compared the performance of LLM-link and GraphRouter across three scenarios, as shown in the first table below. We observed that adding inner edges to LLM nodes impaired the performance of GraphRouter.We suspect that in the settings where all LLMs are observable, the similarities and differences in capabilities of all LLMs have already been captured through extensive message passing via query nodes, task nodes, and their interaction edges. This makes adding these inner edges of LLMs redundant in message passing.

---

> > > ### Author Response · Authors · 2024-11-22
> > > **Response to Reviewer pHun (3/5)**
> > >
> > > **Continuation of the response to Q2:**
> > >
> > > On the other hand, we are very pleased with the reviewer raising this question, as well as Reviewer c8tQ's Q6. We considered that adding inner edges among LLM nodes could enable GraphRouter to address zero-shot scenarios. Specifically, inspired by recommendation systems that often rely on the social networks of new and old users to address the cold start problem of new users, we considered using the addition of inner edges within LLM nodes to enable zero-shot capabilities for new LLMs in GraphRouter. Based on this, we followed the training data settings described in section 4.2, without the need for few-shots data, to conduct zero-shot LLM selection experiments. We set the first six LLMs in Table 3 as visible during training, while the last four are used solely for zero-shot experiments. We connected nodes within the same size or LLM series and conducted zero-shot experiments using this version of GraphRouter in the Balance scenario, as shown in the second table below. We observed that under the zero-shot setting, GraphRouter (zero-shot) not only had an extremely low time cost but also approached the reward of the strongest baseline, C2MAB-V, in this scenario. All these findings demonstrate the great potential of modeling inner links of LLM nodes for the zero-shot capabilities of GraphRouter.
> > >
> > > To conclude, although adding inner edges among LLM nodes does not necessarily enhance GraphRouter’s performance when all LLMs are observable, we have demonstrated that modeling internal connections between LLM nodes is promising to enhance GraphRouter in the zero-shot setting. All the discussions and experimental results mentioned above have been updated in Section D of the Appendix and Table 28-29 of the revised pdf version.
> > >
> > >
> > > **Table 1: Comparison of GraphRouter and LLM-link across different scenarios focusing on Reward**
> > > This table evaluates reward variations between two models under three distinct scenarios. Each number is formatted to three decimal places for precision.
> > >
> > > | Model      | Performance First | Balance | Cost First |
> > > |------------|-------------------|---------|------------|
> > > | GraphRouter| 0.539             | 0.209   | 0.064      |
> > > | LLM-link   | 0.530             | 0.192   | 0.061      |
> > >
> > > **Table 2: Comparison of methods in the zero-shot and few-shot setting on Reward, Time Cost, and the corresponding percentage Reward improvements and Time Cost reduction rate, relative to the most costly method (C2MAB-V (dai2024cost)). The experiment is conducted in the Balance scenario.**
> > >
> > > | **Method**              | **Reward** | **Reward Improvement (%)** | **Time Cost** | **Time Cost Reduction (%)** |
> > > |-------------------------|------------|-----------------------------|---------------|-----------------------------|
> > > | HybridLLM               | 0.01       | -94.71                      | 273.45        | 49.57                       |
> > > | FrugalGPT               | 0.171      | -9.52                       | 63.15         | 88.35                       |
> > > | C2MAB-V                 | 0.189      | 0.00                        | 542.25        | 0.00                        |
> > > | GraphRouter (zero-shot) | 0.182      | -3.7                        | 1.00          | 99.82                       |
> > > | GraphRouter (few-shots) | 0.207      | 9.52                        | 3.00          | 99.45                       |
> > > | GraphRouter (Trained)   | 0.219      | 15.87                       | 30.00         | 94.47                       |
> > >
> > > **[Common prior knowledge]**. We agree that this prior knowledge could potentially enhance the LLM descriptions, thereby improving the performance of LLM selection. However, such prior knowledge is difficult to obtain or requires expert experience, thus limiting the method's automation. Moreover, with the emergence of a large number of new LLMs and updates to existing LLMs, acquiring and utilizing this prior knowledge becomes even more challenging. The core focus of our paper is to model the embeddings of LLMs with varying capabilities more effectively. This is achieved by utilizing the interaction data among tasks, queries, and LLMs, along with the message-passing capabilities of GNNs to capture and differentiate these embeddings. We believe that in the real scenario of massive LLM user interactions, this interaction data can enable GraphRouter to model the descriptions of LLMs more accurately and automatically.

---

> > > > ### Author Response · Authors · 2024-11-22
> > > > **Response to Reviewer pHun (4/5)**
> > > >
> > > > **Q3. Figures 2-4 require further explanation. In Figure 2, what does the notation t represent? For Figures 3 and 4, does "effect" refer to performance, and why do some LLMs have multiple dots? Additionally, there appears to be an unfinished sentence in Line 138.**
> > > >
> > > > **Response:**  Thanks for your insightful feedback. We will answer your questions one by one.
> > > >
> > > > **[Meaning of notation t]**. In fact, we have already clearly introduced in the caption of Figure 2 that $t$ represents the aspect in which the small LLM (LLaMA-3 (7b)) outperforms the large LLM (LLaMA-3 (70b)). However, following the reviewer's suggestion, we further clarify this by revising the caption content to: The probability distribution of a small LLM (LLaMA-3 (7b)) having a better performance value than a large LLM (LLaMA-3 (70b)) by $t$ on the Alpaca dataset, where $t$ means the difference in performance between the small LLM and the large LLM and $t \in [-1, 1]$.
> > > >
> > > > **[Meaning of "effect"]**. The effect refers to performance. To make the y-axes of Figures 3 and 4 clearer, we followed the reviewer's suggestion and changed "effect" to "performance".
> > > >
> > > > **[Multiple dots in Figures 3 and 4]**. We would like to clarify  Figures 3 and 4 represent the distribution of performance across queries from two different datasets for ten different LLMs. This information is explicitly included in the captions of Figures 3 and 4. Specifically, violin plots [1] were used to illustrate this distribution, where the dot in each plot indicates the median performance. To further clarify the principle behind the plotting, we added the following explanation in the caption: "Specifically, we present a violin plot illustrating the performance of ten LLMs of varying sizes and the dot in each distribution is the median performance."
> > > >
> > > > **[Unfinished sentence in Line 138]**. The last sentence of Line 138 is not an unfinished sentence. The entire paragraph is focused on describing Figure 1. Thus, the last sentence of Line 138 also describes Figure 1. However, we follow the reviewer's suggestion and change the last sentence to "We organize the data in a table, as shown on the right of Figure 1" to further facilitate reader understanding.
> > > >
> > > > **[1]** https://en.wikipedia.org/wiki/Violin_plot
> > > >
> > > > **Q4. Can GraphRouter be extended to a training-free version? For example, could a training-free GNN, e.g., SGC or LightGCN, be adapted to perform the link prediction task? I am curious about the capability of a training-free GNN for the LLM-selection task, as well as the percentage of training samples required to achieve satisfactory performance.**
> > > >
> > > > **Response:** Thanks for the reviewer’s insightful questions. Following the reviewer’s advice, we replaced the GNN in GraphRouter with a lightweight SGC to explore its effectiveness in the LLM selection task. Specifically, we conducted experiments in the Balance scenario and compared the performance of SGC across different proportions of the training set, as shown in the table below. We observed that in the absence of any data, the performance of SGC is notably subpar. As the proportion of the training set increases, the performance of SGC improves. When the training data ratio reaches 80%, it achieves 91% of the performance of GraphRouter with a full training set. Further, under a full training set condition, the performance of SGC is very close to that of GraphRouter. These observations validate the potential of the lightweight GNN framework, and we will conduct further research and discussion in future work. All the discussions and experimental results mentioned above have been updated in Section C.4 of the Appendix and Table 27 of the revised pdf version.
> > > >
> > > > **Reward values for SGC corresponding to various training data ratios.**
> > > >
> > > > | **Training Data Ratio** | **0%** | **40%** | **80%** | **100%** |
> > > > |-------------------------|--------|---------|---------|----------|
> > > > | **Reward**              | 0.0692 | 0.154   | 0.19    | 0.203    |

---

> > > > > ### Author Response · Authors · 2024-11-22
> > > > > **Response to Reviewer pHun (5/5)**
> > > > >
> > > > > **Q5. LLM selection task is also dependent on specific querying times. For example, in June 2024, the system may recommend GPT-4o for solving a graph reasoning task, while GPT-o1 might become the best choice at a later date. Can such temporal information be incorporated into the methodology or evaluation?**
> > > > >
> > > > > **Response:** Thanks for the reviewer’s insightful feedback. Actually, GraphRouter has already indirectly taken into account this temporal information. As discussed in the introduction ***[lines 76-94]***, leveraging the powerful inductive capabilities of GNNs enables GraphEval to quickly and effectively generalize to new LLMs. In principle, GraphEval shapes accurate LLM embeddings through the interactions between task-query-LLM and the message passing of GNNs. In fact, the GPT-o1 you mentioned here relative to GPT-4o is a new LLM, and the temporal information here, measuring the differences between GPT-o1 and GPT-4o, can be indirectly reflected through the interactions between the LLM and users. Therefore, once GPT-o1 has some interaction data with users, GraphEval can use this data to model an accurate GPT-o1’s description, thus enabling generalization at the few-shots level. Additionally, as we discussed in the response to Q2, by constructing inner edges between LLM nodes, GraphRouter can zero-shot generalize to new LLMs in such temporal scenarios. Moreover, in real-world applications, like most recommendation systems [1,2,3], our method is live-updated, so any differences in LLMs brought about by this temporal information can be timely incorporated into the model.
> > > > >
> > > > > **[1]** Personalized news recommendation based on click behavior, ICLR 2010.
> > > > >
> > > > > **[2]** Expertise recommender: a flexible recommendation system and architecture, CSCW 2000.
> > > > >
> > > > > **[3]** Getting to know you: learning new user preferences in recommender systems, IUI 2002.

---

> ### Author Response · Authors · 2024-11-24
> **Could you let us know if our rebuttal has sufficiently addressed your concerns?**
>
> Dear Reviewer pHun,
>
> We recognize that the timing of this discussion period may not align perfectly with your schedule, yet we would greatly value the opportunity to continue our dialogue before the deadline approaches.
>
> We hope that our responses and additional experiments have effectively addressed your concerns. We truly appreciate all the valuable advice we have received. Could you let us know if your concerns have been adequately addressed? If you find that your concerns have been resolved, we would appreciate it if you could reconsider the review score.
>
> Thanks!

---

> > ### Comment · Reviewer_pHun · 2024-11-25
> >
> > Thank you for the authors' response. My concerns and questions have been sufficiently addressed.
> >
> > I will maintain my positive score and am more than willing to **champion** this paper.

---

> > > ### Author Response · Authors · 2024-11-25
> > > **Thanks for the Reviewer’s Constructive Feedback**
> > >
> > > Thank you for your thoughtful and constructive feedback. We are pleased to hear that our responses have sufficiently addressed your concerns. We are committed to incorporating the suggested changes in our revisions to further enhance the manuscript. We also appreciate your recognition and support for our paper.

---

### Official Review · Reviewer_c8tQ · 2024-11-04

**Soundness:** 3
**Presentation:** 3
**Contribution:** 3
**Rating:** 6
**Confidence:** 5

**Summary:**

This paper introduces GraphRouter, a novel graph-based approach for selecting appropriate LLMs for different queries. The authors construct a heterogeneous graph comprising task, query, and LLM nodes, with interactions represented as edges to capture contextual relationships. Through an innovative edge prediction mechanism, GraphRouter can adapt to both existing and newly introduced LLMs without requiring retraining. The work demonstrates significant performance improvements over baseline methods across multiple experimental settings, achieving at least 12.3% improvement in standard scenarios and 9.5% improvement in new LLM scenarios. The framework is evaluated on four distinct tasks (Alpaca, GSM8K, SQUAD, Multi-News) using ten different LLMs under various performance-cost tradeoff conditions.

**Strengths:**

The paper seems to be the first to reformulate LLM selection as a graph-based edge prediction problem, providing a fresh perspective on router design. The heterogeneous graph structure effectively captures the complex relationships between tasks, queries, and LLMs. The framework addresses real-world challenges in LLM deployment, particularly the ability to handle new LLMs without retraining and balance performance with computational costs. The evaluation across three different cost-performance scenarios demonstrates practical utility. The authors conduct thorough experiments using multiple datasets, LLMs, and evaluation metrics. The ablation studies on GNN layer count and size provide useful insights for implementation.

**Weaknesses:**

1.	Authors only provide intuitive explanations for why graph structure should help with LLM selection, lacking analysis on why edge prediction correlates with routing performance. Also, the paper fails to explaine why the heterogeneous graph structure (Figure 5) is optimal for capturing LLM-query relationships.
2.	In L. 219, task-query edges are initialized uniformly to 1, which seems overly simplistic given the rich task-query relationships that could be captured. LLM-query edge features only use performance and cost concatenation, ignoring other potentially valuable signals like response length or generation time.
3.	The ablation studies in Section 4.3 don't explore alternative edge feature designs Example: In Table 4, some performance variations could potentially be explained by inadequate edge features, but this is not analyzed.
4.	L. 327 present results across different LLMs but don't analyze how model architectures affect routing decisions. There is no discussion of how to efficiently update the graph structure when new LLMs are added Example: The experiments in Table 5 show impressive few-shot performance but don't evaluate beyond 10 LLMs, leaving questions about larger-scale deployments.
5.	While GNN layer count is analyzed, other hyperparameters and embedding dimension are not thoroughly explored, but I think this is a minor problem.

**Questions:**

1．	How does the performance of GraphRouter change when handling LLMs with similar architectures but different sizes (e.g., different versions of LLaMA)? Is the framework able to effectively distinguish between such similar models?
2．	Could you elaborate on how the framework would handle dynamic updates to LLM capabilities, such as when models are fine-tuned or updated? Would this require rebuilding the entire graph?
3．	Have you considered incorporating more sophisticated edge features, particularly for task-query relationships? How might this affect the framework's performance?

---

> ### Author Response · Authors · 2024-11-22
> **Response to Reviewer c8tQ (1/4)**
>
> **Q1. Authors only provide intuitive explanations for why graph structure should help with LLM selection, lacking analysis on why edge prediction correlates with routing performance. Also, the paper fails to explain why the heterogeneous graph structure (Figure 5) is optimal for capturing LLM-query relationships.**
>
> **Response:** Thanks for your valuable questions and insightful feedback. We answer your questions step by step.
>
> **[Why edge prediction correlates with routing performance]**.  In the paper, we introduced in ***[lines 240-242]*** why edge prediction correlates with routing performance. Specifically, we model the LLM selection problem as predicting and selecting the LLM node with the highest reward for each query node. To achieve this, we construct edge labels by connecting the query node to the LLM node with the highest reward for the training of GraphRouter. This enables the GraphRouter, during the test phase, to select the LLM node connection with the highest probability for each query node as the result of the LLM selection.
>
> **[Design of heterogeneous graph structure]**. In fact, we do not claim that the heterogeneous graph structure is optimal for capturing LLM-query relationships. We believe that our proposed heterogeneous graph structure is sufficiently elegant and naturally models LLM-query relationships, and it has achieved the best results compared to baselines during the experimental process.  Furthermore, in response to Reviewer pHun's Q4, we have explored the use of another lightweight graph structure, such as SGC [1], to address the LLM selection problem. We have included discussions and experimental comparisons of SGC and our graph structure in Section C.4 and Table 27 of the Appendix in our revised PDF.
>
> **[1]** Simplifying graph convolutional networks, ICML 2019.
>
> **Q2. L. 327 presents results across different LLMs but doesn’t analyze how model architectures affect routing decisions. There is no discussion of how to efficiently update the graph structure when new LLMs are added; Example: The experiments in Table 5 show impressive few-shot performance but don't evaluate beyond 10 LLMs, leaving questions about larger-scale deployments.**
>
> **Response:** We thank the reviewer for raising the above concerns. We would like to answer your questions one by one.
>
> **[Graph structure update]**. Indeed, we have introduced the details of updating the graph structure in ***[Figure 5]***. For new queries, LLMs, and tasks, the fundamental update to the graph's structure essentially involves partially adding new edges and nodes. Then, as discussed in the introduction ***[lines 76-94]***,  leveraging the powerful inductive capabilities of GNNs, GraphRouter can efficiently and effectively generalize when faced with updates to the graph.
>
> **[Larger-scale deployments]**. As shown in our related work session, the problem of LLM selection has already been studied by a few existing works (Ding et al., 2024; Chen et al., 2023; Dai et al., 2024). While these works all experiments with fewer than five LLMs, we extend our evaluation with ten LLMs, which is already the largest cale evaluation in this task to date. Moreover, in our experiments, we have utilized all the LLM APIs available for inference on the Together AI platform (some APIs on the platform may have failed).
>
> In addition, the large-scale deployments of GraphRouter are also made possible by its GNN backbone. In established and ordinary GNN tasks, such as those detailed in the Open Graph Benchmark (OGB) [1,2], the number of nodes and edges can escalate to hundreds of millions. On these datasets, GNNs are even capable of training in less than a minute and performing inference in seconds on a standard laptop. In real-world, large-scale deployment scenarios like recommendation systems, GNNs have demonstrated the capability to efficiently manage up to tens of billions of nodes and trillions of edges [3,4]. In contrast, the number of LLM nodes falls short of reaching the million mark, indicating that technically, utilizing GraphRouter on larger-scale scenarios is feasible and efficient without issue. Moreover, in real-world application scenarios, the massive interaction data generated by numerous users enables GraphRouter to learn more refined LLM embeddings, thereby allowing it to better generalize to other LLMs.
>
> **[1]** Open graph benchmark: Datasets for machine learning on graphs, Neurips 2023.
>
> **[2]** Graph neural networks: A review of methods and applications, AI open 2020.
>
> **[3]** Graph convolutional neural networks for web-scale recommender systems, KDD 2020.

---

> ### Author Response · Authors · 2024-11-22
> **Response to Reviewer c8tQ (2/4)**
>
> **Q3. In L. 219, task-query edges are initialized uniformly to 1, which seems overly simplistic given the rich task-query relationships that could be captured. Have you considered incorporating more sophisticated edge features, particularly for task-query relationships? How might this affect the framework's performance? LLM-query edge features only use performance and cost concatenation, ignoring other potentially valuable signals like response length or generation time. The ablation studies in Section 4.3 don't explore alternative edge feature designs Example: In Table 4, some performance variations could potentially be explained by inadequate edge features, but this is not analyzed.**
>
> **Response:** Thanks for the reviewer’s constructive feedback. We answer the questions from two aspects.
>
> **[Embedding similarity-based task-query edges]**.  We have compared the performance of GraphRouter when using more sophisticated edge features versus our current implementation of initializing all edges with a value of 1. To model more complex task-query relationships into the edge, we assign edge values based on the embedding similarity between task descriptions’ embedding and queries embedding, and we name this modified version of our model Edge-similarity. We compared the Reward of GraphRouter and Edge-similarity across three scenarios. As shown in the following table, additional task-query edge information would harm the selection process, as Edge-similarity underperforms GraphRouter across all three scenarios. Although we agree that adapting more advanced relation extraction models could potentially improve the performance, under the current framework, we conclude that complex semantic relationships between queries and tasks would not facilitate the selection performance. Additionally, since this task requires substantial effort and falls outside the primary focus of our paper, we leave this topic for future research. We have concluded these discussions and experiments in section C.2 and Table 25 of the Appendix in our revised PDF.
>
> **Comparison of GraphRouter and Edge-similarity Across Different Scenarios Focusing on Reward.** It evaluates reward variations between two models under three distinct scenarios. Each number is formatted to three decimal places for precision.
>
> | **Model**         | **Performance First** | **Balance** | **Cost First** |
> |-------------------|-----------------------|-------------|----------------|
> | GraphRouter       | 0.539                 | 0.209       | 0.064          |
> |Edge-similarity| 0.510                 | 0.192       | 0.054          |
>
> **[Impact of different LLM-query edge features]** Following the reviewer’s advice, we investigated the Reward of GraphRouter with additional edge features. Specifically, we established three different edge feature combinations, Plus length and Plus time respectively represent the addition of token length and LLM inference time to the GraphRouter base edge features, and Plus length & time represents the addition of both edge features to GraphRouter. As shown in the table below, we compared the Reward values of the origin GraphRouter with the GraphRouter adding the above edge feature combinations across three scenarios.
>
> We observe that while in the Cost First scenarios, an improvement has been shown, it is the least frequently encountered scenario in real-world applications as the actual quality of the LLM’s response is often more prioritized. In the other two scenarios, the gains are minimal or even result in a performance decline. This may be because, in the Performance First and Balance scenarios, Performance significantly contributes to Reward, and both token length and LLM inference time are more closely related to the Cost metric. Therefore, adding these edge features in these scenarios creates certain redundancies, making it difficult to enhance performance. Conversely, in the Cost First scenario, since Cost has a greater impact on Reward, incorporating token length and LLM inference time into the edge features better aids prediction.
>
> Therefore, given the lack of a significant performance boost and the added complexity of incorporating these features, we choose to keep the model simple by only considering the performance and cost features. All the discussions and experimental results mentioned above have been updated in Section C.1 of the Appendix and Table 24 of the revised PDF version.
>
> **Impact on Reward with different edge features.**  We compared the Reward metric under four different combinations of edge features across three scenarios.
>
> | Model             | Performance First | Balance | Cost First |
> |-------------------|-------------------|---------|------------|
> | GraphRouter       | **0.5390**        | 0.2090  | 0.0640     |
> | Plus length       | 0.5283            | 0.2099  | 0.0660     |
> | Plus time         | 0.5245            | 0.2097  | 0.0663     |
> |Plus length & time| 0.5343            | **0.2106**  | **0.0679**     |

---

> > ### Author Response · Authors · 2024-11-22
> > **Response to Reviewer c8tQ (3/4)**
> >
> > **Q4. While GNN layer count is analyzed, other hyperparameters and embedding dimension are not thoroughly explored, but I think this is a minor problem.**
> >
> > **Response:** Thanks for your constructive advice. In the paper, we have already explored the impact of different GNN layer counts and GNN embedding sizes on GraphRouter in section 4.3. However, following the reviewers' suggestions, we have expanded the original ablation study to include experiments on the effects of different training rates on GraphRouter's performance. Specifically, we explored the impact of different learning rates on the Reward of GraphRouter, we selected five different learning rates and compared their effects on Reward in the Balance scenario while keeping other hyperparameters consistent. The table below shows that the Reward generally shows a trend of initially increasing and then decreasing as the learning rate increases, achieving the best performance when the learning rate is 1e-4. All the discussions and experimental results mentioned above have been updated in Section C.3 of the Appendix and Table 26 of the revised PDF version.
> >
> > **Table: Reward values of different learning rates.**
> >
> > | Learning Rate | 1e-1  | 1e-2  | 1e-3  | 1e-4 | 1e-5 |
> > |---------------|-------|-------|-------|------|------|
> > | **Reward**    | 0.0725| 0.0639| 0.0964| 0.208| 0.152|
> >
> > **Q5. How does the performance of GraphRouter change when handling LLMs with similar architectures but different sizes (e.g., different versions of LLaMA)? Is the framework able to effectively distinguish between such similar models?**
> >
> > **Response:** Thanks for the reviewer’s insightful question. Due to our current LLM setting, which includes many LLMs with similar architectures but different sizes (e.g., different versions of LLaMA), our experimental results also indirectly prove GraphRouter's ability to model and distinguish similar LLMs. However, following the reviewers' suggestions, we conducted experiments on different versions of LLaMA. We extracted and utilized data related to the LLaMA series of LLMs from the interaction dataset introduced in section 3.2 for training and prediction. The selected LLMs include LLaMA-3 (7b), LLaMA-2 (7b), LLaMA-3 (70b), LLaMA-3-Turbo (8b), LLaMA-3-Turbo (70b), and Llama-3.1-Turbo (70b). We compared the performance of GraphRouter and the best-performing baseline, FrugalGPT, on this dataset and reported the specific results in the following table. We observed that, compared to FrugalGPT, GraphRouter improved the Reward by at least 10.8%. These observations demonstrate that GraphRouter can effectively capture differences in the capabilities of different LLMs through interactions, achieving good results. All the discussions and experimental results mentioned above have been updated in Section B.2 of the Appendix and Table 23 of the revised PDF version.
> >
> > **Comparison of various methods on the LLaMA-series dataset.** Bold denotes the best results. Each metric reflects average values from multiple evaluation rounds.
> >
> > | Model       |  Performance (Performance First) |  Cost (Performance First) |  Reward (Performance First) |  Performance (Balance) |  Cost (Balance) | Reward (Balance) |  Performance (Cost First) | Cost  (Cost First) |  Reward  (Cost First) |
> > |-------------|--------------------------------|-------------------------|--------------------------|---------------------|--------------|----------------|------------------------|-----------------|-------------------|
> > | FrugalGPT   | 0.382                          | 0.299                   | 0.382                    | 0.367               | 0.043        | 0.162          | 0.372                  | 0.030           | 0.050             |
> > | **GraphRouter** | 0.422                          | 0.307                   | **0.422**                | 0.416               | 0.032        | **0.192**      | 0.416                  | 0.032           | **0.058**         |
> > | Oracle      | 0.489                          | 0.376                   | 0.489                    | 0.459               | 0.051        | 0.204          | 0.436                  | 0.032           | 0.062             |

---

> ### Author Response · Authors · 2024-11-22
> **Response to Reviewer c8tQ (4/4)**
>
> **Q6. Could you elaborate on how the framework would handle dynamic updates to LLM capabilities, such as when models are fine-tuned or updated? Would this require rebuilding the entire graph?**
>
> **Response:** Thanks for your insightful question. When LLM models are fine-tuned or updated, we can view these LLM models as new LLMs. In addition, the strong inductive power enables GraphRouter to effectively learn new LLM nodes' embeddings by modeling their few-shot interactions with task nodes and query nodes. In this process, we can simply add on the previous graph, so there is no need to rebuild the entire graph. The only update required is the new LLM nodes' connections with the few-shots queries. In ***[section 4.2 of our paper and Table 5]***, we demonstrated through experiments that our method generalizes well to new LLMs. Therefore, when models are fine-tuned or updated, GraphRouter can seamlessly adapt by incorporating these changes without the significant computational overhead.
>
> We greatly appreciate you raising this question, as it aligns with a similar point mentioned by Reviewer pHun in Q2. This has also inspired us to explore a method for addressing dynamic updates through a zero-shot approach, requiring only minor modifications to the edges in GraphRouter. Specifically, we draw an analogy to recommendation systems, which rely on the social networks of new and old users to address the cold start problem of new users. In our task, we considered using the addition of inner edges within LLM nodes to enable zero-shot capabilities for new LLMs (dynamically updated) in GraphRouter. We connect LLM nodes within the same size or the same LLM series (such as those within the LLaMA series). Based on this, we followed the training data settings described in section 4.2, but without the need for few-shots data, to conduct zero-shot LLM selection experiments. We set the first six LLMs in Table 3 as visible during training, while the last four are used solely for zero-shot experiments. We connected nodes within the same size or LLM series and conducted zero-shot experiments using this version of GraphRouter in the Balance scenario, as shown in the table below. We observed that under the zero-shot setting, GraphRouter (zero-shot) not only had an extremely low time cost but also approached the reward of the strongest baseline, C2MAB-V, in this scenario. All these findings highlight the great potential of leveraging the inner links of LLM nodes to enhance the zero-shot capabilities of GraphRouter. All the discussions and experimental results mentioned above have been updated in Section D of the Appendix and Table 29 of the revised pdf version.
>
> | **Method**              | **Reward** | **Reward Improvement (%)** | **Time Cost** | **Time Cost Reduction (%)** |
> |-------------------------|------------|-----------------------------|---------------|-----------------------------|
> | HybridLLM               | 0.01       | -94.71                      | 273.45        | 49.57                       |
> | FrugalGPT               | 0.171      | -9.52                       | 63.15         | 88.35                       |
> | C2MAB-V                 | 0.189      | 0.00                        | 542.25        | 0.00                        |
> | GraphRouter (zero-shot) | 0.182      | -3.7                        | 1.00          | 99.82                       |
> | GraphRouter (few-shots) | 0.207      | 9.52                        | 3.00          | 99.45                       |
> | GraphRouter (Trained)   | 0.219      | 15.87                       | 30.00         | 94.47                       |

---

> ### Author Response · Authors · 2024-11-24
> **Could you let us know if our rebuttal has sufficiently addressed your concerns?**
>
> Dear Reviewer c8tQ,
>
> We recognize that the timing of this discussion period may not align perfectly with your schedule, yet we would greatly value the opportunity to continue our dialogue before the deadline approaches.
>
> We hope that our responses and additional experiments have effectively addressed your concerns. We truly appreciate all the valuable advice we have received. Could you let us know if your concerns have been adequately addressed? If you find that your concerns have been resolved, we would appreciate it if you could reconsider the review score.
>
> Thanks!

---

> > ### Author Response · Authors · 2024-11-25
> > **Looking Forward to Further Discussion**
> >
> > Dear Reviewer c8tQ,
> >
> > As the discussion period is drawing to a close, we would like to confirm if our responses have adequately addressed your questions. In light of your valuable feedback, we have provided detailed explanations on the correlation between edge prediction and routing performance, updates and scalability of the graph structure, and the justification for using a heterogeneous graph structure along with its alternatives. Furthermore, we have conducted additional experiments on the impact of embedding similarity-based task-query edges, different LLM-query edge features, an expanded ablation study on learning rates, distinguishing between similar LLM models, and dynamic updates to LLM capabilities. We appreciate your insightful comments and suggestions, which have significantly contributed to the improvement of our paper. We eagerly await your feedback.
> >
> > Best regards,
> >
> > The Authors

---

### Official Review · Reviewer_3cRz · 2024-11-06

**Soundness:** 2
**Presentation:** 2
**Contribution:** 2
**Rating:** 6
**Confidence:** 4

**Summary:**

This paper studies the problem of LLM selection for specific tasks and proposes a graph-based routing framework to select suitable LLM for the input query. By modeling the contextual information among tasks, queries, and LLMs as a heterogeneous graph, query, and LLM nodes, with interactions represented as edges, which efficiently captures the contextual information between the query’s requirements and the LLM’s capabilities. From the experiments, the authors compared the proposed method with different baselines and show that it can outperform regarding the reward. However, there are some issues that need to be addressed for better quality.

**Strengths:**

1. This paper studies the challenging LLM model selection problem, which has been well addressed.

2. This paper considers leveraging graph learning to incorporate more contextual information for LLM model selection.

3. The experiments show that the proposed method achieves good performance compared to baselines.

**Weaknesses:**

1. The studied setting is not quite realistic. The proposed method constructs a graph with task, query, and LLM nodes. For each query, it may select a different LLM to answer the query, which is quite unrealistic in practice.

2. The model performance especially cost may vary a lot on different hardware settings. It is also unrealistic to make sure the real hardware used can align with the numbers in the training data. And we cannot curate the training data for different hardware settings.

3. Some details are unclear. For example, in table 5, what is the specific reward used for evaluation? And what is the purpose of adding task nodes? Can we just incorporate the information of task nodes into the query nodes?

**Questions:**

See weaknesses.

---

> ### Author Response · Authors · 2024-11-22
> **Response to Reviewer 3cRz (1/2)**
>
> **Q1. The studied setting is not quite realistic. The proposed method constructs a graph with task, query, and LLM nodes. For each query, it may select a different LLM to answer the query, which is quite unrealistic in practice.**
>
> **Response:** Thank you for highlighting concerns regarding the studied setting. We answer your questions as follows:
>
> **[LLM selection is realistic in real-world applications]**.  As mentioned in the introduction ***[lines 35-45]*** , as the number of LLMs available rapidly increases, choosing the right LLM to use is also becoming a critical issue.  This claim is supported by the increasing number of LLM API servers, such as Together AI [1] and GroqCloud [3], and these servers usually host multiple different LLM models (usually larger than 10). Moreover, as introduced in section 2.2 and involved in existing research (Ding et al., 2024, Chen et al., 2023, Dai et al., 2024), different LLMs excel at different queries and tasks. Therefore, the selection of LLMs naturally becomes a practical issue to consider,  and designing an efficient LLM selection mechanism capable of addressing various questions and tasks is of practical significance. Furthermore, in the industry, query-level routing is very common. For example, for models, such as ChatGPT [2], which usually serve over 100 million users concurrently, [2], it is impossible to process all user queries on a single model at the same time; it inherently requires a router to distribute queries across multiple LLM models.
>
> **[Modeling LLM selection with graph is natural and useful]**.  As discussed in ***[section 2.2]***, the interactions between tasks, queries, and LLM nodes provide critical contextual information regarding the performance and cost of how LLMs solve different queries and tasks. This heterogeneous contextual information can be naturally modeled as a heterogeneous graph (Peng et al.,
> 2019; Hu et al., 2020). Moreover, in our experiment, GraphRouter substantially surpasses existing routers, delivering a minimum performance improvement of 12.3%. This validates the effectiveness of modeling pass interaction data with heterogeneous graphs, and framing the LLMs selection problem into predicting the most probable edge in the graph.
>
> We hope that the further introduction provided above can clarify the importance of our studied setting. We also kindly anticipate that you, together with the other reviewers who have recognized the significance and authenticity of this issue, will advocate for collective efforts toward its resolution and development.
>
> **[1]** https://docs.together.ai/docs/introduction
>
> **[2]** https://explodingtopics.com/blog/chatgpt-users
>
> **[3]** https://console.groq.com/docs/overview
>
> **Q2. The model performance especially cost may vary a lot on different hardware settings. It is also unrealistic to make sure the real hardware used can align with the numbers in the training data. And we cannot curate the training data for different hardware settings.**
>
> **Response:**  We appreciate that the reviewer pointed out the concerns on model performance. However, we believe that the reviewer might have misunderstood our definitions of model performance and cost. We have also emphasized our definitions in multiple sections of the paper ***[sections 3.2 and 3.4]***. We also follow the settings of existing work (Ding et al., 2024, Chen et al., 2023, Dai et al., 2024), and define model performance as the average quality of the responses under task-specific metrics ***[lines 345-348]***, and cost as the average price of tokens used by the LLM to answer queries ***[lines 348-350]***. Our definitions of model performance and cost do not involve training data for different hardware settings. Moreover, other reviewers have already recognized the reasonableness of our setting, we kindly hope that the reviewer will also acknowledge the significance and practicality of our problems.

---

> > ### Author Response · Authors · 2024-11-22
> > **Response to Reviewer 3cRz (2/2)**
> >
> > **Q3. Some details are unclear. For example, in table 5, what is the specific reward used for evaluation? And what is the purpose of adding task nodes? Can we just incorporate the information of task nodes into the query nodes?**
> >
> > **Response:** Thanks for your valuable questions and insightful feedback. We answer your questions step by step.
> >
> > **[Specific reward used for evaluation]**. We apologize for the confusion. The specific reward used in the evaluation is the same as the reward used in the scenario Balance ***[mentioned in section 4.2, lines 351-359]***, where Performance and Cost have equal importance ***[mentioned in section 3.4, lines 358-366]***.  For a more comprehensive and detailed explanation, we encourage the reviewer to refer to these sections. Additionally, we are happy to provide further clarification if needed.
> >
> > **[Purpose of adding task nodes]**. We believe that adding task nodes has the following advantages over directly incorporating the information of task nodes into the query nodes: **1) It helps us better define the LLM selection problem.** As discussed in section 2.1, the LLM selection process involves interactions among the task, query, and LLMs. Therefore, it is more rational to model the task node separately from a conceptual perspective. **2) It allows the information of the task to play a more significant role in the graph modeling process.** Our analysis in ***[section 2.2 and figures 3 and 4]*** show that different LLMs exhibit notable performance variations across different tasks. This underscores the critical importance of modeling task information in LLM selection. On the other hand, incorporating task information as part of the query node information would diminish the importance of modeling tasks. As a result, it is necessary to add task nodes in our heterogeneous graph to better define the interaction processes. We have also described this intuition in detail in section 2.2. **3) It makes the modeling of graph message passing more flexible and general.** The essence of graph message passing is to update node embeddings by incorporating neighborhood node information, which includes methods like sum, concatenate, and attention (Kipf & Welling, 2017; Hamilton et al., 2017; Veličković et al., 2017). In the context of LLM selection, with task nodes being the neighbors of the query nodes, we can incorporate the information of task nodes into the query nodes and use it to update the embeddings of query nodes. Moreover, directly incorporating the information of task nodes into the query nodes is essentially equivalent to the concatenate operation in message passing, and is a special case of message passing.

---

> > > ### Author Response · Authors · 2024-11-24
> > > **Could you let us know if our rebuttal has sufficiently addressed your concerns?**
> > >
> > > Dear Reviewer 3cRz,
> > >
> > > We recognize that the timing of this discussion period may not align perfectly with your schedule, yet we would greatly value the opportunity to continue our dialogue before the deadline approaches.
> > >
> > > We hope that our responses and additional experiments have effectively addressed your concerns. We truly appreciate all the valuable advice we have received. Could you let us know if your concerns have been adequately addressed? If you find that your concerns have been resolved, we would appreciate it if you could reconsider the review score.
> > >
> > > Thanks!

---

> > > > ### Author Response · Authors · 2024-11-25
> > > > **Looking Forward to Further Discussion**
> > > >
> > > > Dear reviewer 3cRz,
> > > >
> > > > As the discussion period ends soon, we would like to check whether our responses answer your questions. Following your insightful comments, we have clarified our studied setting for LLM selections, model performance, and reward definition, which can be clearly identified in the paper and are highly practical in real-world applications. Additionally, we have illustrated the purpose of adding task nodes. Thank you again for your comments and suggestions to improve our paper, and we look forward to your reply.
> > > >
> > > > Best,
> > > > Authors

---

> > > > > ### Comment · Reviewer_3cRz · 2024-11-27
> > > > >
> > > > > Thanks for your response and most of my concerns have been addressed. I would like to change my rating to 6.

---

> > > > > > ### Author Response · Authors · 2024-11-28
> > > > > > **Thanks for the Reviewer’s Constructive Feedback**
> > > > > >
> > > > > > Thank you for your thoughtful and constructive feedback. We are pleased to hear that our responses have addressed most of your concerns. We are committed to incorporating the suggested changes in our revisions to further enhance the manuscript.

---

### Meta-Review · Area_Chair_1qhU · 2024-12-17

**Metareview:**

This paper proposes GraphRouter, a graph-based routing system which is used to select appropriate LLMs to handle input queries.  The authors propose a graph ML angle on modeling the context information between queries, tasks, and LLMs as a heterogeneous graph, with interactions as edges.  The authors introduce a reward metric which offsets cost from performance  and demonstrate their proposal outperforms others in terms of net reward.

Overall, reviewers consistently evaluated this work positively.  There were a few themes of comments which I encourage the authors to fold into the revision:

- Improved clarifications about the introduction of various graph metadata (e.g task nodes as 3cRz raised, graph definition choices which c8tQ raised, node features as z8Nx raised)

- Discussion and rationale around the practicality of the evaluation setting and proposed approach (pHun, 3cRz, z8Nx) -- some reviewers still have doubts on this point and some further text in the manuscript to justify the choices made would be helpful.

Despite these limitations, the work proposes an interesting interplay conceptually between graph ML models and their use in the space of LLM selection.

**Additional Comments On Reviewer Discussion:**

Authors addressed several of the above concerns during the rebuttal.  In particular, they introduced new experiments around the impact on reward with different choices of edge features including embedding similarity-based ones (in response to c8tQ).  Moreover, the authors introduced new experiments on "dynamically updated" LLMs as raised by c8tQ and pHun to demonstrate good zero/few-shot performance.  Also, the authors introduced several new LLM backbones as well as new tasks (e.g. code generation) in response to simplified evaluation setting concerns from pHun.

---

### Decision · Program_Chairs · 2025-01-22

Accept (Poster)